# Double burden of malnutrition and associated factors among adolescent in Ethiopia: A systematic review and meta-analysis

**Aragaw Gezaw** [1]*, **Wolde Melese**[1], **Bekalu Getachew**[1], **Tefera Belachew**[2]

1 Department of Public Health Nutrition, School of Public Health, College of Health Science, Wollo University, Dessie, Ethiopia, 2 Faculty of Public Health, Department of Biomedical Science, Department of Nutrition and Dietetics, Jimma University, Jimma, Ethiopia

* aragawgezaw@gmail.com

**Data Availability Statement:** All relevant data are within the paper.

**Funding:** The author(s) received no specific funding for this work.

## Abstract

### Background

As adolescence is a transition period from childhood to adulthood malnutrition occurring at this age resonates through generations. Although there were many individual studies in Ethiopia about different form of malnutrition among adolescent, their results are inconclusive indicating the need for generating a pooled estimate of adolescent nutritional status and associated factors. This review and meta-analyses aimed at estimating the pooled prevalence of different forms of malnutrition and associated factors among adolescents in Ethiopia.

### Method and materials

We searched data bases from Pub Med, Cochrane Library, Health Inter Network Access to Research Initiative (HINARI), Science Direct and search engines; Google and Google Scholar and other sources; Reference of References and expert contact which were used to select the studies. Joanna Briggs Institute (JBI) quality appraisal tool was applied to identify eligible studies. STATA/SE V.14 was used to analyze the data. Effect size with 95% Confidence Interval (CI) and heterogeneity were estimated. Heterogeneity of studies was quantified with $I^2$ statistic >50% used as an indicator of heterogeneity. Potential publication bias was assessed using Funnel plots and Egger's regression test. Trim and fill analysis was also performed. The presences of a statistical association between independent and dependent variables were declared at $P < 0.05$. The PROSPERO registration number for the review is CRD42020159734.

### Results

The pooled prevalence of overweight/obesity, stunting and thinness were 10.63% (95% CI: 8.86, 12.40), 20.06% (95% CI: 15.61, 24.51) and 21.68% (95% CI: 9.56, 33.81), respectively. Being female (OR: 2.02, CI: 1.22–3.34), low dietary diversity score (OR: 2.26 CI: 1.28–3.99) and high physical activity (OR: 0.36, 95%CI: 0.14–0.88) were significantly

**Competing interests:** The authors have declared that no competing interests exist.

**Abbreviations:** BMI, Body Mass Index; CI, Confidence Interval; DDS, Dietary Diversity Score; JBI, Joanna Briggs Institution; MeSH, Medical Subject Headings; OR, Odds Ratio; PRISMA, Preferred Reporting Items for Systematic Review and Meta-Analysis; SDG, Sustainable Development Goal; SE, Standard Error; WHO, World Health Organization.

associated with adolescent overweight/obesity. Urban residence (OR: 0.82, 95%CI: 0.68–0.99), protected drinking water source (OR: 0.50, CI: 0.27–0.90) and having family size<5 people (OR: 0.54, CI: 0.44–0.66) were independent predictors of adolescent stunting. Early adolescent age (10–14 years) (OR: 2.38, CI: 1.70–3.34), protected water source for drinking (OR: 0.36, CI: 0.21–0.61), low wealth index (OR: 1.80, CI: 1.01–3.19) and family size <5 people (OR: 0.50, CI: 0.28–0.89) were significantly (P < 0.05) associated with adolescent thinness.

## Conclusion

The prevalence of overweight/obesity, stunting and thinness are high in Ethiopian adolescents indicating the upcoming challenge of double burden of malnutrition. The results imply the presence of double burden of malnutrition among adolescents which heralds the need for programmatic and policy response in terms of addressing modifiable risk factors including: dietary practices, physical activity, water source and economic status of these adolescents.

## Introduction

Adolescence period is characterized by physical, biological, psychological and social maturation signifying the transition to adulthood. According to World Health Organization (WHO) adolescents represent in the age group of 10–19 years [1]. Adolescents represent almost 20% of the world population and approximately 84% are living in developing countries. Sub- Saharan African adolescents make 23% of the population. In Ethiopia they represent 20–26% of the population [1–7]. According to WHO, interest and focus on adolescent health and nutrition is relatively recent. Conversely, the rate of growth during adolescence is the second fastest next to growth that occurs in the first 1000 days of life implying the need for direct nutrition intervention for this age-group. Although adolescents need a continuum care from childhood through adolescent, they are often ignored [1,8–10].

Previously the focus of nutrition agenda in low and middle-income countries has been on under-nutrition. Rapid economic development and urbanization have given rise to a nutrition transition, where energy-dense foods replace traditional foods and sedentary lifestyles prevail leading to an increase in obesity and diet-related chronic non-communicable diseases. Coexistence of under nutrition and over nutrition poses a public health challenge [11–14]. Therefore, it is imperative to find ways to eliminate under-nutrition and its associated morbidity and mortality, without contributing to obesity and risk of nutrition-related chronic diseases.

Malnutrition is deviations from the optimum body needs (it can be undernutrition or overnutrition). It is mainly caused by unbalanced, inadequate, or excessive intake of nutrients. Undernutrition refers to insufficient intake of dietary energy and nutrients that fulfill the body's demand for optimum function. Undernutrition manifests in the form of stunting or wasting/thinness. Stunting is a chronic form of undernutrition which is caused by inadequate nutrition over a long period that fails to attain optimum growth while wasting/thinness is an acute form of undernutrition that indicates a recent food shortage and/or infectious diseases that leads to rapid and severe weight loss [15–19]. Overnutrition include overweight and obesity which is abnormal or excessive accumulation of fat that may result in health impairment [18–20].

To assess the nutrition status of adolescents, the WHO currently recommends using BMI-for-age and height-for-age. Thinness (low body-mass-index (BMI)-for-age (BAZ)) z-score is

below minus 2 (-2.0) standard deviations (SD) and stunting (low height-for-age (HAZ)) z-score is below minus 2 (-2.0) standard deviations (SD). Overweight is (high Body Mass Index (BMI)-for-age)) greater than plus 1 standard deviation; and obesity is greater than plus 2 standard deviations above the WHO Growth Reference median [17–19,21].

Optimal nutrition during adolescence is a prerequisite for proper physical, mental, and social development. During adolescence; boys can achieve a linear growth of 9.5 cm/year while girls can increase 8.3 cm per year [22–25]. They can also increase in weight as much as half of their adult body weight. This rapid growth can be taken as a window of opportunity compensating for early childhood growth failure. Adolescent period is not only a time for tremendous growth, but also time of considerable risk. Suboptimal nutrition during adolescence results in delayed pubertal development, delayed sexual maturation, low lean body mass accretion, slower linear growth and future adverse health outcomes (adult physique and sense of self-esteem, metabolic and cardiovascular problems) and productivity [1,8,26–28].

Ethiopia has been implementing different strategies and programs to ensure food and nutrition security, as part of its national development agenda such as the Food Security Strategy, National Nutrition Strategy, National Nutrition Program, the Seqota Declaration roadmap, Nutrition Sensitive Agriculture Strategy, School Health and Nutrition Strategy and the Productive Safety Net Program through multi-sectoral nutrition coordination and integration [29,30]. However, health services in Ethiopia are not meeting the need of adolescents, instead focusing on preschool children and pregnant women, which resulted in lack of attention to adolescents and left several questions unanswered. Absolute nutrient requirements are higher during adolescence compared to childhood due to increased growth and body size [31]. Adolescent boys have greater requirements for most nutrients compared to girls due to differences in growth and development [20].

There is also lack of consistent information regarding factors associated with adolescent nutritional status; the difference in sex, age, residence, household wealth index, and lifestyle including alcohol and tobacco use, eating habits, level of physical activity, parent educational status, family size and sanitation.

Quantifying the double burden of malnutrition among adolescents and examining the associated factors across the country are important for policymakers to support actions for achievement of the Sustainable Development Goal (SDG) of ending malnutrition in all its forms by 2030. Nutrition is an indispensable cog without which the SDG machine cannot function smoothly [32–34]. Health care providers should have up-to-date and state-of-the-art evidence on adolescent malnutrition and associated factors to provide integrated nutrition services. Therefore, this study aimed at determining the pooled prevalence malnutruition and associated factors among Ethiopian adolescents.

## Materials and methods

This study followed the recommended statement of Preferred Reporting Items for Systematic Reviews and Meta-Analyses (PRISMA) [35]. The procedures of screening and selection of eligible studies were presented using the PRISMA flow diagram [36]. The protocol for this systematic review and meta-analysis was registered at the international prospective register of systematic review and meta-analysis (PROSPERO) with a registration ID of CRD42020159734.

### Eligibility criteria

**Inclusion criteria.** All studies reported on malnutrition and associated factors among adolescents in Ethiopia (both male and female between 10–19 years), observational studies

including descriptive cross-sectional, analytical cross-sectional, case–control and cohort studies were included, all articles regardless of publication status but reported only in the English language were included. All forms of overnutrition (overweight/obesity) or undernutrition (stunting and thinness) with their prevalence and associated factors done in Ethiopia were also considered without time restriction.

**Exclusion criteria.** Studies conducted among special population (such as adolescents living with HIV/AIDS, tuberculosis) and mental disorders were excluded.

## Information sources and search strategy

PROSPERO registrations as well as databases were explored to confirm whether previous systematic review and/or meta-analysis exist in order to avoid duplicates. To access published primary studies, PubMed/Medline, Cochran library, Science Direct and Health Inter Network Access to Research Initiative (HINARI) databases were used. Grey literature was retrieved using Google and Google Scholar searching engines. The reference lists of the retrieved studies were probed through contact relevant experts and organization's website to collect articles that were not accessible through databases and search engines.

The search was restricted to only 'human studies' and 'English language' to suppress the number of irrelevant studies in the advanced search. The following were the key search terms: "nutritional status", "overnutrition", "overweight", "obesity", "undernutrition", "stunting", "thinness", "wasting", "malnutrition", "determinants*", "associated factors", "adolescents" and "Ethiopia". In the advanced search of databases, 'Medical Subject Headings (MeSH)' terms and to linking 'All fields' "AND" and "OR" Boolean operator and "*" truncation was used as appropriate (Table 1).

**Table 1. Search strategies of adolescent malnutrition and associated factors in Ethiopia.**

| Data bases | Search strategy |
|---|---|
| Pubmed | (((("nutritional status"[Mesh] OR "malnutrition"[Mesh] OR "under nutrition"[Mesh] OR "over nutrition"[Mesh] OR stunting[Mesh] OR thinness[Mesh] OR overweight[Mesh] OR obesity [Mesh] OR nutrition*[tiab])) AND ((Adolescent[Mesh] OR youth[Mesh] OR teenager[Mesh] OR adolescen*(tiab) OR youth*(tiab) OR teen*(tiab)))) AND (("Ethiopia"[MeSH Terms] OR "Ethiopia"[All Fields])) |
| HINNARI | (((("nutritional status"[Mesh] OR "malnutrition"[Mesh] OR "under nutrition"[Mesh] OR "over nutrition"[Mesh] OR stunting[Mesh] OR OR thinness[Mesh] OR overweight[Mesh] OR obesity [Mesh] OR nutrition*[tiab])) AND ((Adolescent[Mesh] OR youth[Mesh] OR teenager[Mesh] OR adolescen*(tiab) OR youth*(tiab) OR teen*(tiab)))) AND (("Ethiopia"[MeSH Terms] OR "Ethiopia"[All Fields])) |
| Cochrane library | ((((((((((("nutritional status") OR undernutrition) OR thinness) OR stunting) OR overweight) OR obesity) OR overnutrition) OR malnutrition) AND AND adolescent) AND Ethiopia |
| Science direct | "Nutritional status"[Mesh] OR undernutrition OR stunting OR wasting OR overnutrition OR overweight OR obesity OR malnutrition |
| **Search engines** | |
| Google | (((("nutritional status"[Mesh] OR "malnutrition"[Mesh] OR "under nutrition"[Mesh] OR "over nutrition"[Mesh] OR stunting[Mesh] OR thinness[Mesh] OR overweight[Mesh] OR obesity [Mesh] OR nutrition*[tiab])) AND ((Adolescent[Mesh] OR youth[Mesh] OR teenager[Mesh] OR adolescen*(tiab) OR youth*(tiab) OR teen*(tiab)))) AND (("Ethiopia"[MeSH Terms] OR "Ethiopia"[All Fields])) |
| Google scholar | (((("nutritional status"[Mesh] OR "malnutrition"[Mesh] OR "under nutrition"[Mesh] OR "over nutrition"[Mesh] OR stunting[Mesh] OR thinness[Mesh] OR overweight[Mesh] OR obesity [Mesh] OR nutrition*[tiab])) AND ((Adolescent[Mesh] OR youth[Mesh] OR teenager[Mesh] OR adolescen*(tiab) OR youth*(tiab) OR teen*(tiab)))) AND (("Ethiopia"[MeSH Terms] OR "Ethiopia"[All Fields])) |

## Study selection

Articles gathered from different sources were exported to Endnote X7, and duplicates were identified and removed. The remaining articles were evaluated in the context of the topic, study participants, language and study area. The independent reviewers screened the title and abstract of each study. Agreement between the two reviewers were accepted when Cohen's kappa coefficient was>0.6. The screening was repeated when the kappa value is <0.6 [37]. After reaching good agreement, a full-text review was performed. When there was disagreement the review resolved through discussion.

## Data collection process

After eligible studies were identified, two independent reviewer (AG and WM) extract the relevant data using a standardized data extraction format, which was adopted from the JBI data extraction format prepared on Microsoft Excel spreadsheet [38] and edited by(TB). Pilot test was conducted for all data extraction forms using a representative sample of the studies to be reviewed and the form was corrected based on pilot test finding. Finally, the two reviewers independently extracted the full texts of the name of the first author, year of publication, region, residence, study design, sex, age, sample size, number of nutritional status(overweight/ obesity, stunted and thinness/ wasted), response rate, study quality score and raw data of associated factors on a 2x2 table using the structured data extraction format.

## Quality appraisal and risk of bias in individual studies

The quality assessment was performed by two independent reviewers (AG and WM). The quality of each article was assessed using the standardized Joanna Briggs Institute(JBI) critical appraisal tool prepared for case–control, analytical cross-sectional and descriptive cross-sectional studies each with 10, 8 and 9 question items was assessed respectively [38]. All tools have 'Yes' and 'No' types of questions and scores were given 1 for 'Yes' and 0 for 'No' responses. Scores were sum up and transformed into percentage. Only studies that scored $\geq$50% were considered for both systematic review and meta-analysis of prevalence [39]. For any scoring disagreements, which were happening between the assessors, the source of discrepancy was investigated through making a thorough revision and disagreements were resolved through discussion. The quality results of primary studies were placed in a separate column of the data extraction format.

## Operational definition

Overweight is BMI for age Z-score of more than +1 standard deviations (SD) from the median of the reference population and obesity is BMI-for-age Z-score of more than +2 SD. Adolescents whose height-for-age Z-score is below minus two (-2 SD) from the median of the reference population are considered short for their age (stunted). Underweight 'is weight for age z-score < -2SD.

## Summary measures

Effect measures were calculated for each study. Since, the data are dichotomous, effect size was calculated for prevalence and odds ratio (OR). The values of ratio measure, odds ratio underwent log transformations before being analyzed. The log transformed values made the scale symmetric to perform the analyses, and then converted the results back to ratio values for interpretation. Summary effect was different in the two models. In the fixed-effect analysis we assumed that the true effect size was the same in all studies, and the summary effect was our

estimate of this common effect size. In the random-effects analysis we assume that the true effect size varies from one study to the next, and that the studies in our analysis represent a random sample of effect sizes that could have been observed. The summary effect is our estimate of the mean of these effects. The precision which encompassed three formal statistics, the variance, standard error, and confidence interval that addresses the accuracy of the summary effect as an estimate of the true effect were determined.

## Synthesis of results

The extracted data were exported to STATA/SE V.14 for further analysis. The data were synthesized based on nutritional status (both over and under nutrition). The existence of heterogeneity among studies was examined by Forest plot, Cochran's Q statistics ($X^2$ test) and the size of $I^2$. The $I^2$ values of 25%, 50% and 75% were interpreted as the presence of low, medium and high heterogeneity, respectively [40]. A pooled estimate of the prevalence was generated for the different form of malnutrition (overweight/obesity, stunting and thinness) separately among adolescents. Potential cause of heterogeneity was explored by meta-regression analysis, sensitivity analysis and subgroup analyses.

Results were presented in narratives, tables and forest plots. All statistical interpretation was reported based on p-value and 95% CI. The presence of a statistical association between independent and dependent variables was declared based on p-value of <0.05. Finally, the findings of the qualitative studies were combined, and an integrative approach of quantitative–qualitative meta-synthesis was carried out.

## Risk of bias across studies

Publication bias was assessed using funnel plot subjectively and Egger's test objectively (35, 36). Publication bias was adjusted by trim (remove unmatched observation) and fill (imputing values for missed studies) methods of analysis.

## Ethics and dissemination

Ethical clearance was obtained from School of Public Health, College of Medicine and Health Sciences, Wollo University. Even if ethical clearance was not required for this review as primary data were not collected, Ethical clearance was given for corresponding author(s) via mail and other means of communication for articles with full texts that were hard to access. The copy of this systematic review and meta-analysis was given to Wollo University.

# Results

## Study selection

In the initial search, a total of 1289 records were found from different electronic search databases, search engine and other sources: Pub Med (38), Cochrane Library (56), HINARI (1090), Science Direct (85) and search engines; Google (5), Google Scholar (1), other sources; cross references (13) and expert contact (1). From these, 114 duplicate records were removed and 1112 records were excluded after screening by title and abstracts. We assessed the full texts of 63 remaining records for eligibility, and 30 records were further excluded by the exclusion criteria. Finally, 33 studies were considered for the final review and meta-analysis [1,4,41–71] (Fig 1). Of the 33 studies, 14 studies for overweight/obesity, 25 studies for stunting, and 25 studies for thinness were used to estimate the pooled prevalence of adolescent nutritional status (Fig 1).

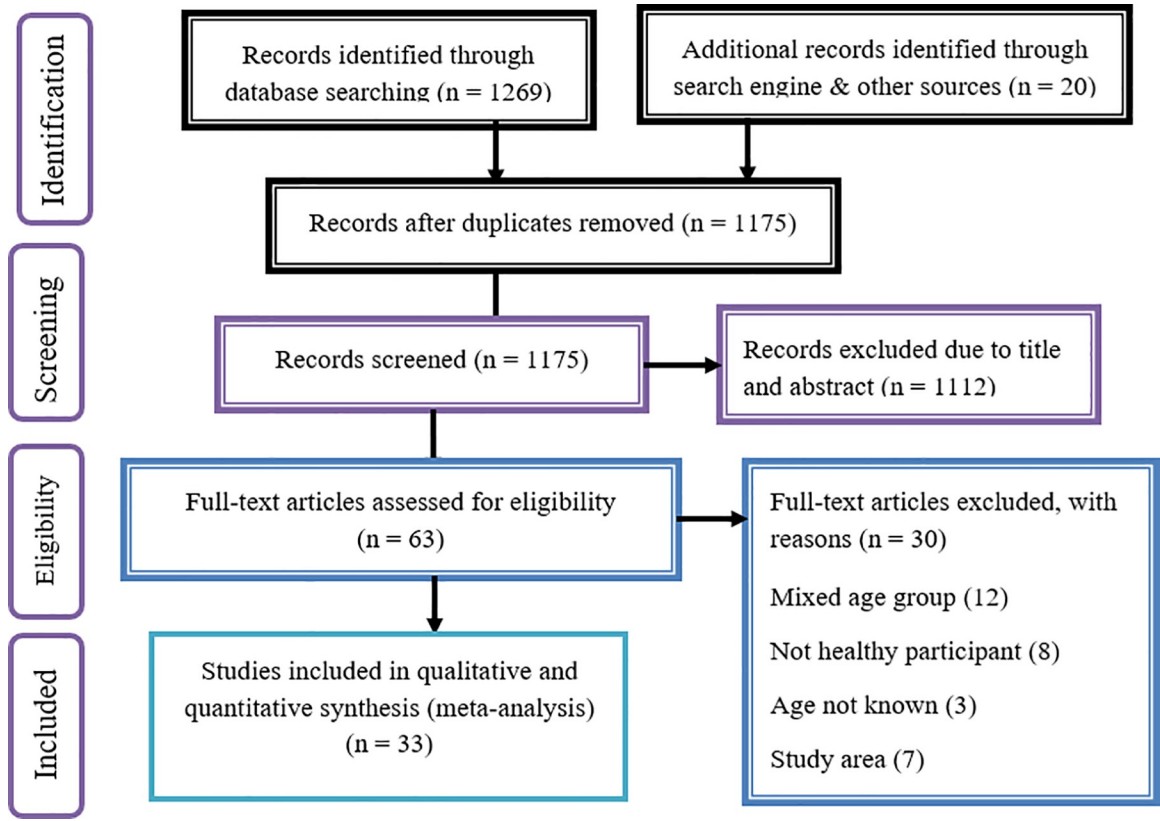

**Fig 1. PRISMA 2009 flow diagram of study selection of adolescent double burden of malnutrition in Ethiopia.**

## Characteristics of the studies and systematic review

The studies included in this systematic review and meta-analysis were 30 descriptive cross-sectional studies, 2 comparative cross sectional studies and one case control study. A total of 25,172 adolescents were included in the analysis. The included studies reported sample size ranging from 174 [65] to 2733 [41]. All included studies were conducted among female participants [1,4,41–71], but for overweight/obesity 12 studies were conducted among both male and female [4,43,46,48,51,53–56,65,70], for thinness 13 studies were conducted among both male and female [1,4,42,45,46,48,49,53,54,56,68,70,71] and for stunting 12 studies were conducted among both male and female participants[1,42,45,48,49,54,56–58,67,68,71]. The pooled estimate of overweight/obesity was high among females, while thinness was high among male participants which were 13% and 28%, respectively (Table 2). Eighteen (54.5%) of the included studies were conducted in both urban and rural areas [1,4,41,44–49,52,56–58,62,65,67,71]. Among all included studies, two were conducted in Addis Ababa[43,54], one at the national level [41], one in Afar Region [59], 6 in Amhara Region [44,49,62,66,67,71], 11 in Oromia Region [45,48,51,56–58,60,63,65,69,70], 5 in SNNP [4,46,55,61,64], 2 in Somalia Region [42,50] and 5 in Tigray Region [1,47,52,53,68] (Table 3).

Quality appraisal was performed for 33 studies using JBI critical appraisal tool. The quality score ranged from 80–100% and all of them were included. Regarding the respective study design; quality score ranged from 89–100% for 30 descriptive cross-sectional studies, for two comparative cross-sectional studies quality score was 89 and 100%, while for case control studies a quality score of 80% was observed (S1 Table).

**Table 2. Summary of extracted studies subgroup by sex in the analysis of prevalence and associated factors of adolescent's malnutrition in Ethiopia.**

| S. No | Author | Publication year | Region | Female obese | Total Female | Male obese | Total male | stunted female | Total female | Stunted male | Total male | Female Thin | Total Female | Male Thin | Total male |
|---|---|---|---|---|---|---|---|---|---|---|---|---|---|---|---|
| 1 | Abate B, et al | 2020 | Ethiopia | NR | NR | NR | NR | 380 | 2733 | NR | NR | NR | NR | NR | NR |
| 2 | Abdulkadir A et al | 206 | Somalia | NR | NR | NR | NR | 29 | 347 | 46 | 308 | 78 | 308 | 72 | 347 |
| 3 | Alemu E, et al | 2014 | AA | 12 | 211 | 63 | 589 | NR | NR | NR | NR | NR | NR | NR | NR |
| 4 | Arage G. et al | 2019 | Amhara | NR | NR | NR | NR | 59 | 362 | NR | NR | 105 | 362 | NR | NR |
| 5 | Assefa et al | 2015 | Oromo | NR | NR | NR | NR | 103 | 942 | 210 | 1014 | 839 | 941 | 738 | 1010 |
| 6 | Berbada, D et al | 2017 | SNNP | 31 | 324 | 36 | 276 | NR | NR | NR | NR | 52 | 324 | 66 | 276 |
| 7 | Berhe, K.et al | 2020 | Tigray | NR | NR | NR | NR | 133 | 398 | NR | NR | 128 | 398 | NR | NR |
| 8 | Damie, T et al | 2015 | Oromo | 11 | 106 | 1 | 185 | 11 | 106 | 10 | 185 | 11 | 106 | 60 | 185 |
| 9 | Demilew, Y et al | 2018 | Amhara | NR | NR | NR | NR | 33 | 219 | 68 | 188 | 3 | 165 | 26 | 242 |
| 10 | Engidaw, M. et al | 2019 | Somalia | NR | NR | NR | NR | 45 | 415 | NR | NR | 63 | 415 | NR | NR |
| 11 | Gali, N.et al | 2017 | Oromo | 53 | 303 | 15 | 207 | NR | NR | NR | NR | NR | NR | NR | NR |
| 12 | Gebregyorgis, T. et al | 2016 | Tigray | NR | NR | NR | NR | 99 | 814 | NR | NR | 174 | 814 | NR | NR |
| 13 | Gebremariam, H et al | 2015 | Tigray | 7 | 290 | 6 | 265 | NR | NR | NR | NR | 78 | 290 | 132 | 265 |
| 14 | Gebreyohannes, Y et al | 2014 | AA | 57 | 503 | 30 | 521 | 34 | 503 | 40 | 521 | 13 | 503 | 51 | 521 |
| 15 | Geta, M et al | 2017 | SNNP | 57 | 171 | 43 | 129 | NR | NR | NR | NR | NR | NR | NR | NR |
| 16 | Hassen, K. Etal | 2017 | Oromo | 28 | 312 | 11 | 238 | 50 | 312 | 36 | 238 | 34 | 312 | 30 | 238 |
| 17 | Irenso, A.et al | 2020 | Oromo | NR | NR | NR | NR | 224 | 982 | 317 | 1028 | NR | NR | NR | NR |
| 18 | Jikamo, B. et al | 2019 | Oromo | NR | NR | NR | NR | 330 | 985 | 214 | 973 | NR | NR | NR | NR |
| 19 | Kahssay, M. et al | 2020 | Afar | NR | NR | NR | NR | 78 | 340 | NR | NR | 30 | 310 | NR | NR |
| 20 | Kt, Roba et al | 2016 | Oromo | 28 | 700 | NR | NR | 109 | 700 | NR | NR | 149 | 700 | NR | NR |
| 21 | melaku, Y. A. et al | 2015 | Tigray | NR | NR | NR | NR | 58 | 154 | 41 | 194 | 41 | 194 | 50 | 154 |
| 22 | Roba, A. C.et al | 2015 | SNNP | 26 | 188 | NR | NR | 58 | 188 | NR | NR | 3 | 185 | NR | NR |
| 23 | Tariku, A. et al | 2019 | Amhara | NR | NR | NR | NR | 734 | 1150 | NR | NR | NR | NR | NR | NR |
| 24 | Teferi, D. et al | 2018 | SNNP | 31 | 315 | 3 | 340 | 34 | 655 | NR | NR | 7 | 315 | 25 | 340 |
| 25 | Tegegne, M, etal | 2016 | Oromo | NR | NR | NR | NR | 71 | 598 | NR | NR | 125 | 473 | NR | NR |
| 26 | Teshome, T et al | 2013 | SNNP | 55 | 274 | 16 | 280 | NR | NR | NR | NR | NR | NR | NR | NR |
| 27 | Wakayo, T et al | 2016 | Oromo | 16 | 99 | 2 | 75 | NR | NR | NR | NR | NR | NR | NR | NR |
| 28 | Wassie, M et al | 2015 | Amhara | NR | NR | NR | NR | 404 | 1281 | NR | NR | 170 | 1252 | NR | NR |
| 29 | Woday, A et al | 2018 | Amhara | NR | NR | NR | NR | 33 | 277 | 47 | 238 | NR | NR | NR | NR |
| 30 | Yebyo H et al | 2015 | Tigray | NR | NR | NR | NR | 45 | 208 | 60 | 203 | 70 | 208 | 111 | 203 |
| 31 | Yemaneh, Y et al | 2012 | Oromo | NR | NR | NR | NR | 130 | 642 | NR | NR | 95 | 642 | NR | NR |
| 32 | Yetubie, M.et al | 2010 | Oromo | 9 | 183 | 9 | 242 | NR | NR | NR | NR | 45 | 183 | 72 | 242 |
| 33 | Zemene, M et al | 2019 | Amhara | NR | NR | NR | NR | 32 | 167 | 17 | 160 | 11 | 167 | 5 | 160 |

NR = Not Reported.

## Prevalence of adolescent double burden of malnutrition

Thirteen studies were included to estimate the pooled prevalence of overweight/obesity [4,43,46,48,51,53,54,56,60,61,64,65,70]. Heterogeneity among the studies was used to estimate the pooled prevalence of adolescent overweight/ obesity was low ($I^2 = 31.4\%$, P = 0.132). Using fixed-effects model (Mantel-Haenszel), the pooled prevalence of adolescent overweight/obesity was 10.6% (95% CI: 8.86, 12.40) (Fig 2). Twenty five studies were included in the analysis to

Table 3. Summary of extracted studies in the analysis of prevalence and associated factors of adolescent's malnutrition in Ethiopia.

| S. No. | Author | Publication Year | Region | Residence | Study Design | Sex | Age | Sample size | Response Rate | Overweight/ Obesity | Stunting | Thinness | Quality Score |
|---|---|---|---|---|---|---|---|---|---|---|---|---|---|
| | | | | | | | | | (%) | (%) | (%) | (%) | (%) |
| 1 | Abate B, et al | 2020 | Ethiopia | Both | CS | Female | 15–19 | 2733 | 78 | NR | 15 | NR | 89 |
| 2 | Abdulkadir A et al | 2016 | Somalia | Unclass. | CCS | Both | 10–19 | 655 | 100 | NR | 11.45 | 22.90 | 88 |
| 3 | Alemu E, et al | 2014 | AA | Urban | CS | Both | 15–19 | 800 | 100 | 9.38 | NR | NR | 100 |
| 4 | Arage G. et al | 2019 | Amhara | Both | CS | Female | 10–19 | 362 | 100 | NR | 16.30 | 29.01 | 89 |
| 5 | Assefa et al | 2015 | Oromo | Both | CS | Both | 10–19 | 1956 | 93.8 | NR | 16.00 | 80.62 | 100 |
| 6 | Berbada, D et al | 2017 | SNNP | Both | CS | Both | 10–19 | 600 | 94.6 | 11.17 | NR | 19.67 | 100 |
| 7 | Berhe, K.et al | 2020 | Tigray | Both | CS | Female | 10–19 | 398 | 100 | 32.16 | 32.16 | 32.16 | 89 |
| 8 | Damie, T et al | 2015 | Oromo | Both | CS | Both | 10–19 | 291 | 91.2 | 4.12 | 5.84 | 24.40 | 100 |
| 9 | Demilew, Y et al | 2018 | Amhara | Both | CS | Both | 15–19 | 407 | 95.9 | NR | 24.82 | 7.13 | 100 |
| 10 | Engidaw, M. et al | 2019 | Somalia | Unclass. | CS | Female | 10–19 | 415 | 98.1 | NR | 10.84 | 15.18 | 100 |
| 11 | Gali, N.et al | 2017 | Oromo | Unclass. | CS | Both | 10–19 | 510 | 93.4 | 13.33 | NR | NR | 100 |
| 12 | Gebregyorgis, T. et al | 2016 | Tigray | Both | CS | Female | 10–19 | 814 | 98.9 | NR | 12.16 | 21.38 | 100 |
| 13 | Gebremariam, H et al | 2015 | Tigray | Unclass. | CS | Both | 10–19 | 555 | 97 | 2.34 | NR | 37.84 | 100 |
| 14 | Gebreyohannes, Y et al | 2014 | AA | Unclass. | CCS | Both | 13–19 | 1024 | 100 | 8.50 | 7.23 | 6.15 | 100 |
| 15 | Geta, M et al | 2017 | SNNP | Unclass. | CC | Both | 12–15 | 300 | 100 | 33.33 | NR | NR | 80 |
| 16 | Hassen, K. Etal | 2017 | Oromo | Both | CS | Both | 10–19 | 550 | 96 | 7.09 | 15.64 | 11.64 | 100 |
| 17 | Irenso, A.et al | 2020 | Oromo | Both | CS | Both | 10–19 | 2010 | 100 | NR | 2.69 | NR | 100 |
| 18 | Jikamo, B. et al | 2019 | Oromo | Both | CS | Both | 13–17 | 2084 | 100 | NR | 26.10 | 25.29 | 100 |
| 19 | Kahssay, M. et al | 2020 | Afar | Unclass. | CS | Female | 10–19 | 340 | 97.7 | NR | 22.94 | 8.82 | 100 |
| 20 | Kt, Roba et al | 2016 | Oromo | Urban | CS | Female | 13–19 | 700 | 97.2 | 3.00 | 15.57 | 21.29 | 89 |
| 21 | melaku, Y. A.et al | 2015 | Tigray | Both | CS | Both | 10–19 | 348 | 100 | NR | 28.45 | 26.15 | 100 |
| 22 | Roba, A. C.et al | 2015 | SNNP | Unclass. | CS | Female | 15–19 | 188 | 100 | 13.83 | 30.85 | 1.60 | 89 |
| 23 | Tariku, A. et al | 2019 | Amhara | Both | CS | Female | 10–19 | 1550 | 100 | NR | 47.35 | NR | 100 |
| 24 | Teferi, D. et al | 2018 | SNNP | Both | CS | Both | 10–19 | 655 | 95.2 | 5.19 | 5.19 | 4.89 | 100 |
| 25 | Tegegne, M, etal | 2016 | Oromo | Both | CS | Female | 10–19 | 598 | 96.9 | NR | 20.90 | 11.87 | 100 |
| 26 | Teshome, T et al | 2013 | SNNP | Urban | CS | Both | 10–19 | 554 | 93 | 15.52 | NR | NR | 100 |

(*Continued*)

**Table 3.** (Continued)

| S. No. | Author | Publication Year | Region | Residence | Study Design | Sex | Age | Sample size | Response Rate (%) | Overweight/ Obesity (%) | Stunting (%) | Thinness (%) | Quality Score (%) |
|---|---|---|---|---|---|---|---|---|---|---|---|---|---|
| 27 | Wakayo, T et al | 2016 | Oromo | Both | CS | Both | 11–18 | 174 | 98 | 10.92 | NR | 18.97 | 100 |
| 28 | Wassie, M et al | 2015 | Amhara | Rural | CS | Female | 10–19 | 1281 | 97.2 | NR | 31.54 | 13.27 | 100 |
| 29 | Woday, A et al | 2018 | Amhara | Both | CS | Both | 10–19 | 515 | 96.26 | NR | 15.53 | NR | 100 |
| 30 | Yebyo H et al | 2015 | Tigray | Unclass. | CS | Both | 10–19 | 411 | 100 | NR | 26.28 | 44.04 | 100 |
| 31 | Yemaneh, Y et al | 2012 | Oromo | Rural | CS | Female | 10–19 | 642 | 94 | NR | 20.25 | 14.80 | 100 |
| 32 | Yetubie, M.et al | 2010 | Oromo | Rural | CS | Both | 10–19 | 425 | 100 | 4.24 | NR | 27.53 | 100 |
| 33 | Zemene, M et al | 2019 | Amhara | Both | CS | Both | 15–19 | 327 | 93.69 | NR | 14.98 | 4.89 | 100 |
| | TOTAL | | | | | | | 25172 | | | | | |

CS = Cross-sectional

CCS = Comparative Cross-sectional

CC = Case Control

unclass. = unclassified

NR = Not Reported.

estimate the pooled prevalence of adolescent stunting [1,4,41,42,44,45,47–50,52,54,56–63,66–69,71]. The heterogeneity among the 25 studies used to estimate the pooled prevalence of adolescent stunting was high ($I^2$ = 97.0%, P < 0.001). Using the random-effects model, the overall pooled prevalence of adolescent stunting was 20.1% (95% CI: 15.61, 24.51) (Fig 3).

Twenty-five studies were included to estimate the pooled prevalence of thinness [1,4,42,44–50,52–54,56,58–61,63,65,66,68–71]. Heterogeneity among the studies used to estimate the pooled prevalence of adolescent thinness was high ($I^2$ = 99.7%, P < 0.001). The pooled prevalence of adolescent thinness was 21.7% (95% CI: 9.56, 33.81) using random-effects model (Fig 4).

**Subgroup analysis for prevalence of adolescent double burden of malnutrition.** Since there was high heterogeneity on the pooled effect of studies conducted on stunting and thinness, subgroup analysis were performed to identify the possible source of heterogeneity. The pooled prevalence estimated for adolescent stunting was high in Amhara Region (25.2%, 95% CI: 13.24, 37.11) and Tigray Region (25.4%, 95% CI: 18.36, 32.40), and the least was in Somalia (11.2% (95%CI: 7.24, 15.07). Heterogeneity was high except in Somalia Region (88.8–99.1%). Subgroup analysis by gender showed that comparable pooled estimate of stunting were (19.9%, 95%CI: 14.81–25.66) and (20.2% 95% CI: 12.47–27.96) among males and females, respectively. Heterogeneity was high (92.9%) among both sexes. High heterogeneity was also observed in other subgroups; residence, age, study design and sample size (Table 4).

The pooled estimate of adolescent thinness was high in Tigray Region (32.4, 95%CI: 24.63, 40.16) and least in Addis Ababa and Afar (7.7, 95%CI: 2.88, 12.61). Heterogeneity was high except in Addis Ababa and Afar Region 84.6–99.8%. Another subgroup analysis showed that adolescent thinness was high in sample size of ≥384 (23.3%, 95%CI: 8.35, 38.19). Subgroup analysis by gender showed that pooled estimate of thinness was high among males (27.9% 95%

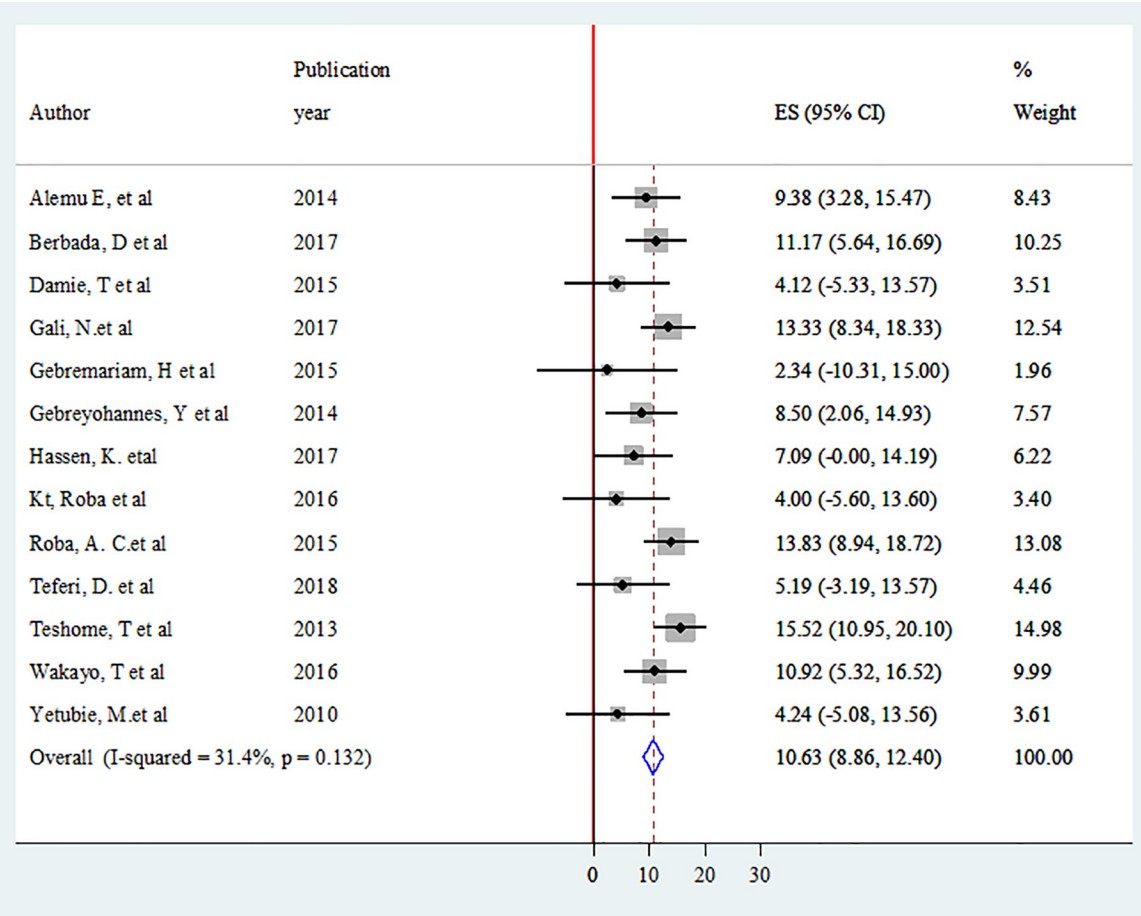

**Fig 2. Forest plot of the pooled prevalence of adolescent overweight/obesity in Ethiopia.**

CI: 15.33–40.52). Heterogeneity was high among other subgroups; sex, age, residence, study design and sample size which was 90.3–99.8% (Table 4).

Even if subgroup analysis was performed, source of heterogeneity was not detected. Therefore sensitivity analysis and meta-regression were considered to identify the source of heterogeneity.

**Sensitivity analysis for prevalence of adolescent double burden of malnutrition.** Sensitivity analyses were performed for effect sizes of all of the studies on stunting and thinness to identify the possible source of heterogeneity and to single out the effect of one study on the overall estimate. However none of studies were found to be neither showed a statistically significant source of heterogeneity nor a significant influence in all the analysis (Figs 5 and 6).

**Meta-regression for prevalence of adolescent double burden of malnutrition.** Meta regression was done for each variable to identify the possible source of heterogeneity in the pooled estimate of stunting and thinness. Univariate meta-regression analysis was done by taking publication year of the studies and the sample size to detect the potential source(s) of variation. Sample size for studies on thinness was significant (p = 0.031) indicating that sample size is the source of heterogeneity. Binary meta-regression was undertaken for other binary variables, but none of them were found to be statistically significant source of heterogeneity (Table 5).

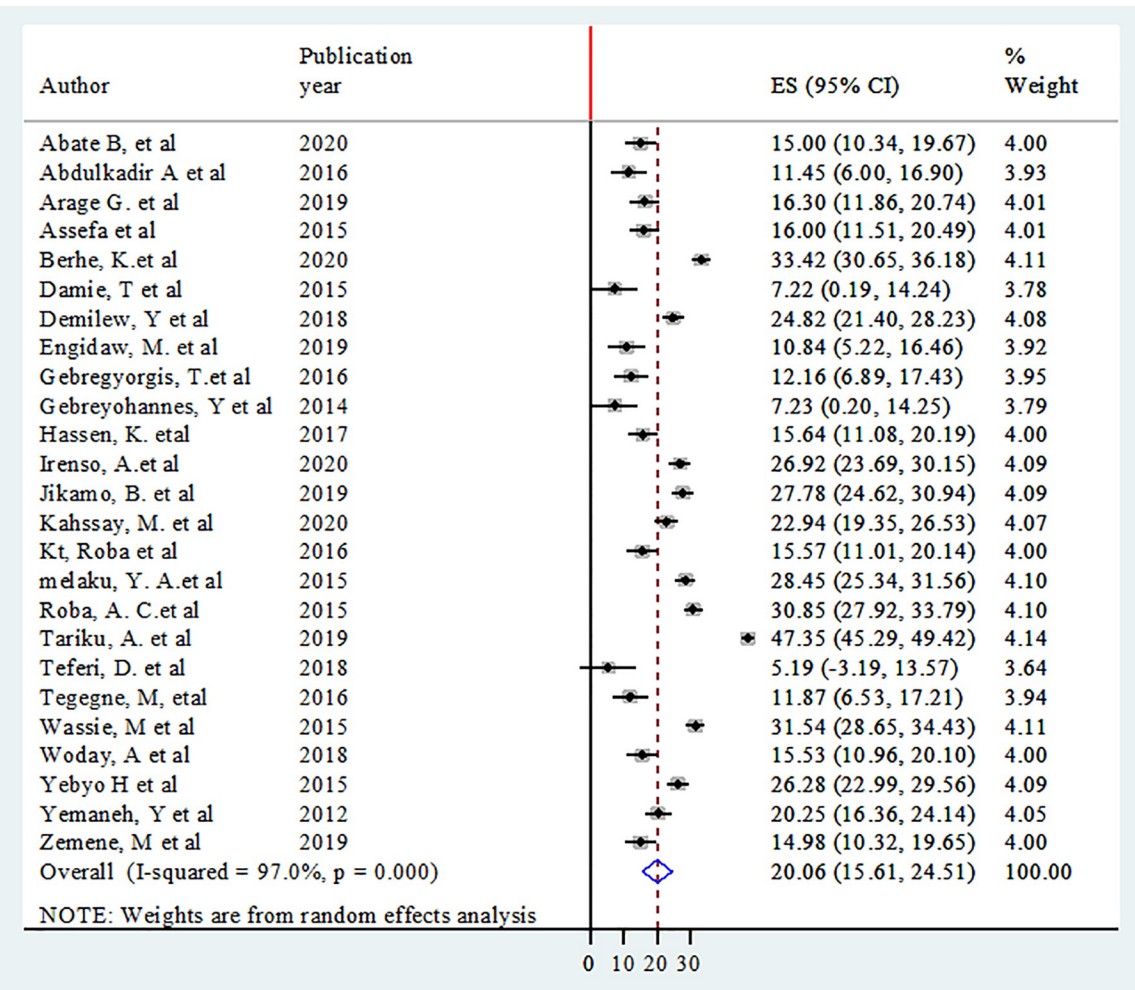

**Fig 3. Forest plot of the pooled prevalence of adolescent stunting in Ethiopia.**

**Publication bias for prevalence of adolescent double burden of malnutrition.** Possible publication bias was subjectively examined using funnel plot and objectively determined using Egger's test at 5% significant level. The funnel plots were asymmetrical and Eggers' regression test (p<0.001) were significant. Both funnel plot and Eggers' test results showed that there is a significant publication bias (Figs 7–9). Therefore, the final effect size was determined by applying Trim and Fill analysis. However, there was no change in effect size (Figs10–12).

## Factors associated with double burden of adolescent malnutrition

Eight factors were identified for overweight/obesity that could be used in the quantitative meta-analysis. Weights were calculated using the random-effects analysis since heterogeneity was observed among all factors. Statistical association were not observed among adolescent age, residence, wealth index, meal frequency and family size <5. Female adolescents had almost 2 times higher odds of being overweight/ obese compared males (OR: 2.02, CI: 1.22–3.34). Adolescents who had low DDS were 2 times more likely to be overweight/obese (OR: 2.26 CI: 1.28–3.99).Adolescents with high physical activity had 64% lower odds of being overweight/obese (OR: 0.36, 95%CI: 0.14–0.88) (Fig 13).

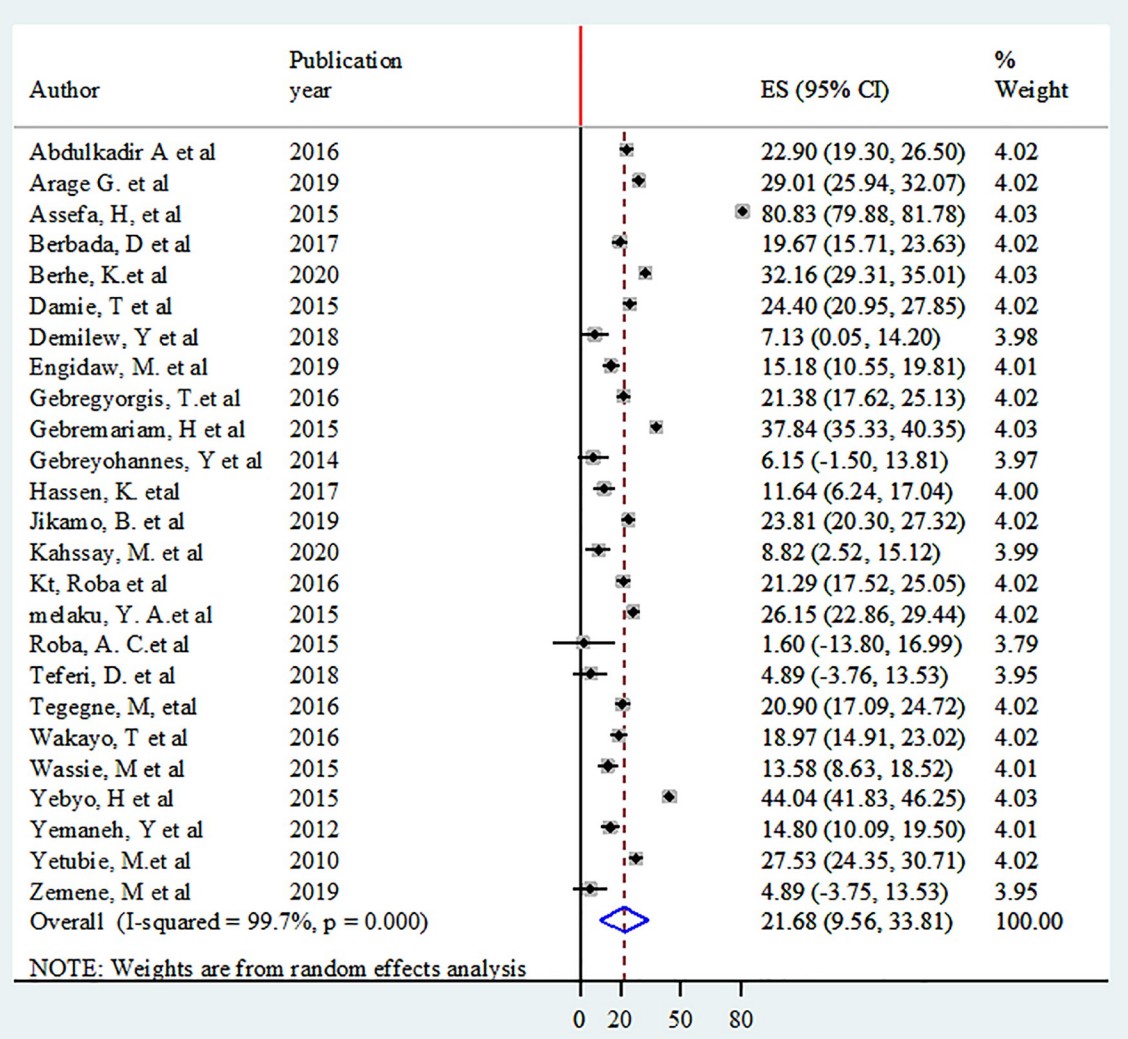

**Fig 4. Forest plot of the pooled prevalence of adolescent thinness in Ethiopia.**

For stunting, six associated factors were used in the quantitative analysis. Although heterogeneity was not observed in family size and residence, it was observed among adolescent age, sex, meal frequency and protected drinking water source. Weights were calculated using the random-effects analysis. Adolescent age, sex, and meal frequency were not statistically significant factors for stunting. Urban adolescences had 18% lower odds of being stunted (OR: 0.82, 95%CI: 0.68–0.99). Adolescents having protected water source for drinking had 50% lower likelihood of stunting (OR: 0.50, CI: 0.27–0.90). Households with family size of < 5 were 46% less likely to be stunted than those with family size of ≥5 (OR: 0.54, CI: 0.44–0.66) (Fig 14).

Heterogeneity was observed in all the seven factors associated with thinness included in the analysis. Thus, weights were calculated using the random-effects analysis. Sex, residence, meal frequency and DDS were not statistically significant. Early adolescents (age 10-14years) were 2 times more likely to be thin than late adolescent (OR: 2.38, CI: 1.70–3.34). Adolescents with protected water source were 64% less likely to be thin than those adolescents had unprotected water source (OR: 0.36, CI: 0.21–0.61). Likewise, households with a family size of <5 were 50% less likely to be thin than those who had family size of ≥5 (OR: 0.50, CI: 0.28–0.89).

Table 4. Subgroup analysis of double burden of malnutrition among adolescent in Ethiopia.

| Sub-group | Variables | Number of studies | Prevalence (95% CI) | Heterogeneity | |
|---|---|---|---|---|---|
| | | | | I²% | P |
| **Overweight/Obesity** | | | | | |
| **Region** | Adis Ababa | 2 | 8.96(4.54–13.38) | 0.0 | 0.846 |
| | SNNP | 4 | 12.88(10.18–15.59) | 40.4 | 0.169 |
| | Oromo | 6 | 9.26(6.44–12.09) | 24.9 | 0.248 |
| | Overall | 13 | 10.63(8.86,12.40) | 31.4 | 0.132 |
| **Sex** | Female | 13 | 13.23(11.65–14.80) | 67.2 | <0.001 |
| | Male | 11 | 8.06(5.57–10.54) | 0.0 | 0.541 |
| | Overall | 13 | 11.75(10.42,13.08) | 59.9 | <0.001 |
| **Stunting** | | | | | |
| **Region** | Other | 3 | 15.49(7.02–23.95) | 88.8 | <0.001 |
| | Somalia | 2 | 11.16(7.24–15.07) | 0.0 | 0.879 |
| | Amhara | 6 | 25.18(13.24–37.11) | 98.7 | <0.001 |
| | Oromo | 8 | 19.02(15.02–23.02) | 86.7 | <0.001 |
| | Tigray | 4 | 25.38(18.36–32.40) | 99.1 | <0.001 |
| | SNNP | 2 | 18.33(-6.81–43.47) | 95.0 | <0.001 |
| | Overall | 25 | 20.69(15.11–26.28) | 97.5 | <0.001 |
| **Sex** | Female | 25 | 20.21(12.47–27.96) | 99.1 | <0.001 |
| | Male | 12 | 19.90(14.81–25.66) | 95.0 | <0.001 |
| | Overall | 25 | 20.03(14.41,24.99) | 98.8 | <0.001 |
| **Age** | 15–19 | 7 | 19.59(13.87–25.30) | 92.9 | <0.001 |
| | 10–19 | 18 | 20.69(15.11–26.28) | 97.5 | <0.001 |
| | Overall | 25 | 20.06(15.61,24.51) | 97.0 | <0.001 |
| **Study design** | DCS | 23 | 21.29(16.84–25.73) | 97.0 | <0.001 |
| | CCS | 2 | 9.86(5.56–14.17) | 0.0 | 0.352 |
| | Overall | 25 | 20.06(15.61,24.51) | 97.0 | <0.001 |
| **Residence** | Urban and rural | 16 | 20.58(14.42–26.74) | 97.4 | <0.001 |
| | Unclassified | 6 | 18.66(11.56–25.76) | 94.2 | <0.001 |
| | Urban | 1 | 15.57(11.01–20.14) | - | - |
| | Rural | 2 | 25.97(14.91–37.03) | 95.2 | <0.001 |
| | Overall | 25 | 20.06(15.61,24.51) | 97.0 | <0.001 |
| **Sample size** | ≥384 | 19 | 20.33(14.89–25.77) | 97.4 | <0.001 |
| | <384 | 6 | 20.53(14.23–26.83) | 93.5 | <0.001 |
| | Overall | 25 | 20.06(15.61,24.51) | 97.0 | <0.001 |
| **Thinness** | | | | | |
| **Region** | Somalia | 2 | 19.18(11.62–26.74) | 85.0 | 0.010 |
| | Amhara | 4 | 13.99(1.69–26.31) | 95.2 | <0.001 |
| | Oromo | 9 | 27.15(2.30–52.00) | 99.8 | <0.001 |
| | SNNP | 3 | 9.93(-2.60–22.47) | 84.6 | 0.002 |
| | Tigray | 5 | 32.40(24.63–40.16) | 97.4 | <0.001 |
| | Other | 2 | 7.74(2.88–12.61) | 0.0 | 0.597 |
| | Overall | 25 | 21.68(9.56,33.81) | 99.7 | <0.001 |
| **Sex** | Female | 23 | 19.94(2.86–37.02) | 99.8 | <0.001 |
| | Male | 13 | 27.93(15.33–40.52) | 99.6 | <0.001 |
| | Overall | 25 | 22.84(12.16,33.51) | 99.7 | <0.001 |
| **Age** | 15–19 | 6 | 12.19(4.29–20.09) | 90.3 | <0.001 |
| | 10–19 | 19 | 24.53(10.48–38.60) | 99.8 | <0.001 |

(*Continued*)

**Table 4.** (Continued)

| Sub-group | Variables | Number of studies | Prevalence (95% CI) | Heterogeneity | |
|---|---|---|---|---|---|
| | | | | I$^2$% | P |
| | Overall | 25 | 21.68(9.56,33.81) | 99.7 | <0.001 |
| **Study design** | CCS | 2 | 14.88(-1.52–31.28) | 93.4 | <0.001 |
| | DCS | 23 | 21.96(9.18–34.74) | 99.7 | <0.001 |
| | Overall | 25 | 21.68(9.56,33.81) | 99.7 | <0.001 |
| **Residence** | Unclassified | 7 | 20.31(25.94–31.28) | 98.1 | 0.001 |
| | Urban and rural | 14 | 22.79(4.04–41.54) | 99.8 | <0.001 |
| | Urban | 1 | 21.29(17.52–25.06) | - | - |
| | Rural | 3 | 18.68(8.87–28.49) | 93.7 | <0.001 |
| | Overall | 25 | 21.68(9.56,33.81) | 99.7 | <0.001 |
| **Sample size** | ≥384 | 18 | 23.27(8.35–38.19) | 99.8 | <0.001 |
| | <384 | 7 | 18.09(12.25–23.94 | 91.0 | <0.001 |
| | Overall | 25 | 21.68(9.56,33.81) | 99.7 | <0.001 |

Adolescents in the households with lower wealth index were almost 2 times more likely to be thin than those in high wealth index (OR: 1.80, CI: 1.01–3.19) (Fig 15).

**Subgroup analysis of factors associated with double burden of adolescent malnutrition.**   Among factors associated with nutritional status (overweight/obesity, stunting and thinness) high heterogeneity was observed in pooled effect of most factors. Subgroup analysis was conducted to explore the possible source of heterogeneity on the overall odds ratio of each factors associated with nutritional status among adolescent by considering potentially important factors. The heterogeneity still persisted in the subgroups of factors associated with respective nutritional status (Table 6). Then, further sensitivity analysis and meta-regression analysis were performed to identify source of heterogeneity.

**Sensitivity analysis of factors associated with double burden of adolescent malnutrition.**   Sensitivity analysis was performed for included factors to identify source of heterogeneity on overall odds ratio of factors associated with nutritional status (overweight, stunting and thinness). But there was no study that showed a significant influence in all the analysis (Figs 16–18).

**Meta-regression of factors associated with double burden of malnutrition.**   Due to the presence of high heterogeneity on pooled odds ratio of factors associated with respective nutritional status, (overweight / obesity, stunting and thinness) of adolescent, meta-regression was conducted to identify the possible cause of heterogeneity in factors associated with nutritional status (overweight/obesity, stunting and thinness). But none of the included variables showed a statistically significant source of heterogeneity in all the analysis (Table 7).

**Publication bias for factors associated with double burden of adolescent malnutrition.**   Potential publication bias among factors associated with nutritional status was examined subjectively (Funnel plot) and objectively (Eggers' test) which showed symmetrical funnel plot and non-significant Egger's test result. Therefore no publication bias was detected among factors associated with nutritional status (overweight/obesity, stunting and thinness) (Figs 19–21).

## Discussion

Malnutrition affects adolescent's sexual maturation and growth, increases the risk of poor obstetric outcomes, affects ability to learn and work with maximum productivity, prevents the

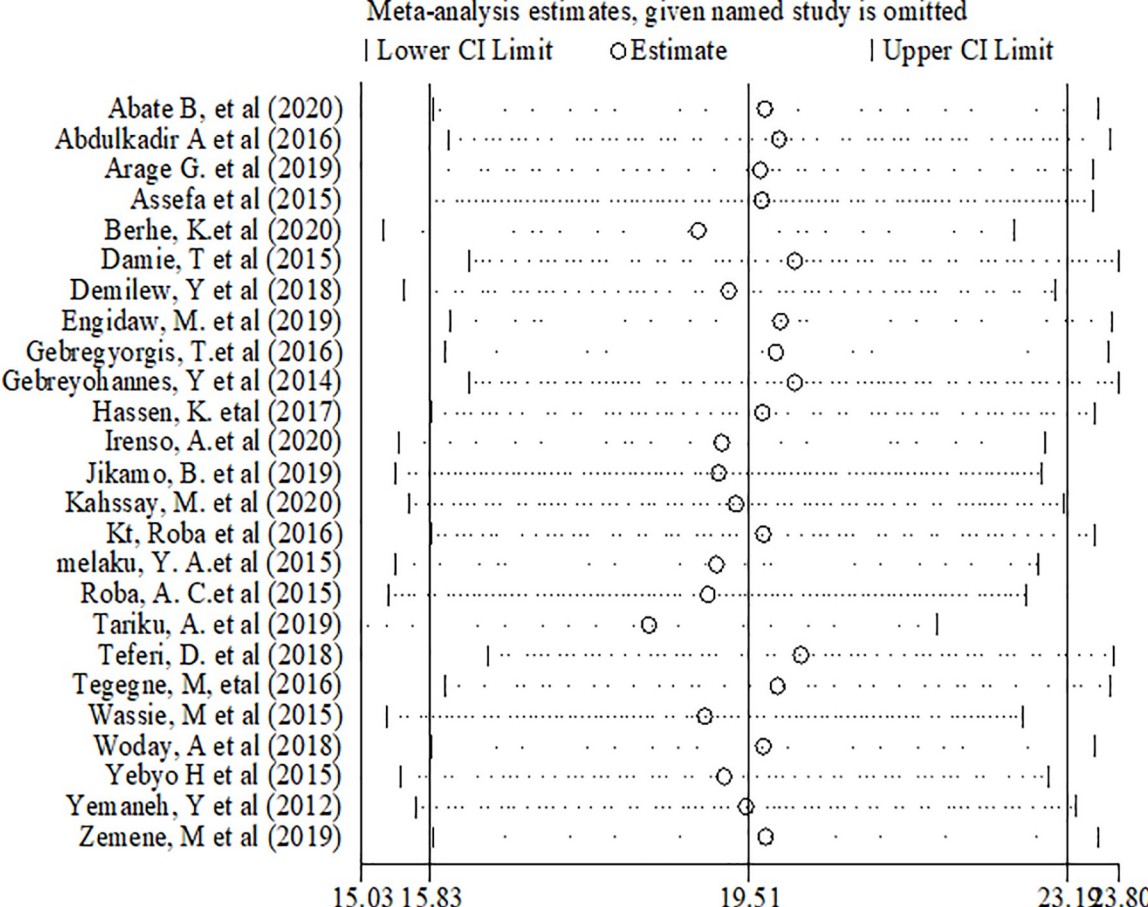

**Fig 5. Sensitivity analysis of stunting among adolescents in Ethiopia.**

attainment of normal bone and teeth strength, increase the risk of chronic disease resonating through generations [25]. Overweight/obesity is an emerging nutritional problem in developing countries, which increases the burden of nutrition-linked non-communicable diseases and has far-reaching consequences on economic growth of countries over time [25,72]. Contrary to previous concerns that mainly focused on the difficulty of undernutrition, over nutrition is currently becoming and emerging nutrition related public health problem. The problem is a double whammy debilitating low income communities in different parts of Africa most importantly in Ethiopia [73]. This systematic review and meta-analysis was conducted to estimate the pooled prevalence and associated factors of adolescent nutritional status in Ethiopia.

The result of this meta-analysis showed that the combined overall prevalence of overweight/obesity was 10.63% (95% CI: 8.86, 12.40%) among adolescents in the country which was higher than overweight/obesity estimated in EDHS; 2005, 2011 and 2016 which was 3, 2.8 and 4%, respectively [74–76]. Although it was cross-sectional study, it serves as a baseline for the country level policies and recommendations. The result was also higher than the study conducted among African school learner adolescent(7%) [77]. The possible reason for this discrepancy might be due to difference in nutritional pattern and physical activities and socio-cultural reasons indicating the increase in overweight/obesity over time. The prevalence of overweight/obesity in this study was less than the finding reported among adolescents in Asia

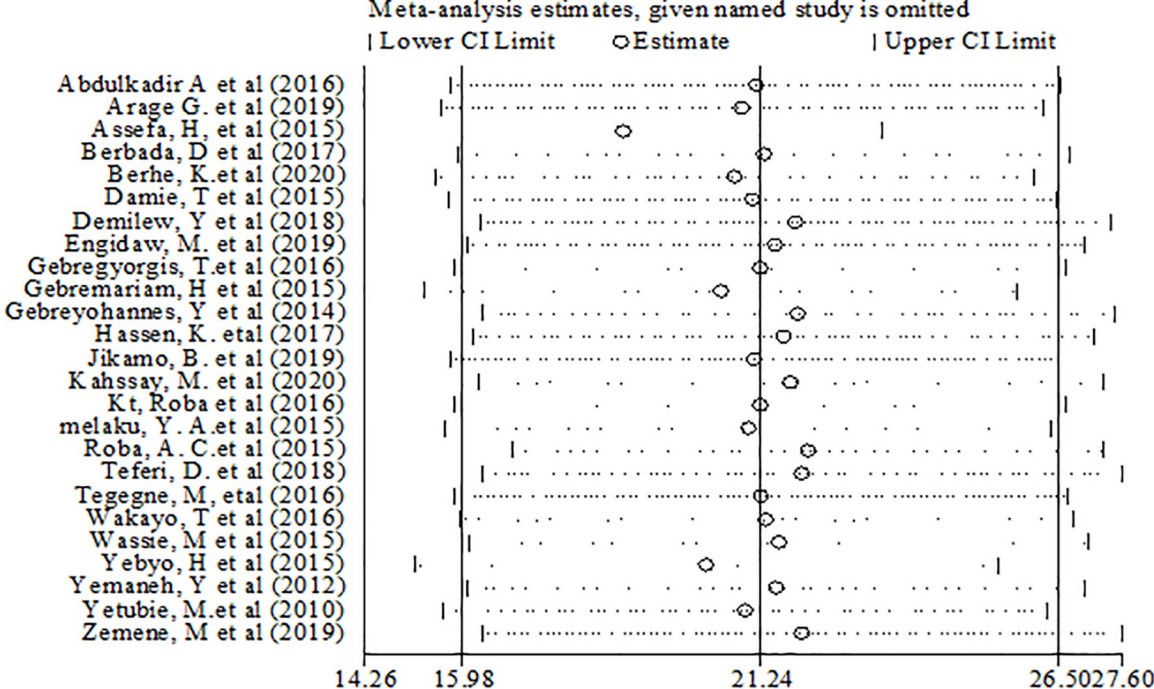

**Fig 6. Sensitivity analysis of thinness among adolescents in Ethiopia.**

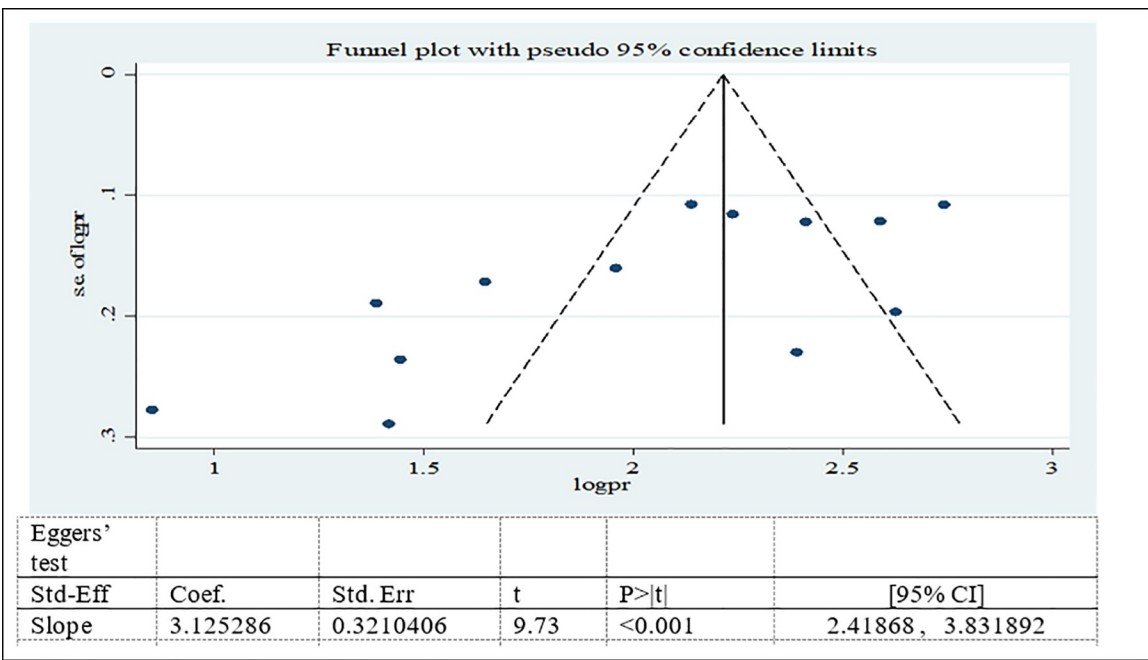

**Fig 7. Funnel plot and Eggers' test to assess publication bias for adolescent overweight/obesity in Ethiopia.**

**Table 5. Meta-regression for factors related to heterogeneity of adolescent malnutrition in Ethiopia.**

| Stunting | | | Thinness | | |
|---|---|---|---|---|---|
| Variables | Coefficients | P | Variables | Coefficients | P |
| Publication year | 0.6113396 | 0.522 | Publication year | -1.149887 | 0.376 |
| Sample size | 0.0015933 | 0.609 | Sample size | 0.015184 | 0.031 |
| **Region** | | | **Region** | | |
| Amhara | 14.151 | 0.093 | Amhara | -5.266813 | 0.703 |
| Oromo | 7.595355 | 0.341 | Oromo | 7.38093 | 0.553 |
| Others* | 4.162138 | 0.648 | Others* | -11.55734 | 0.553 |
| SNNP | 7.908809 | 0.434 | SNNP | 0–9.764877 | 0.509 |
| Tigray | 14.0995 | 0.112 | Tigray | 13.28234 | 0.322 |
| Somalia | Reference | 1 | Somalia | Reference | 1 |
| **Residence** | | | **Residence** | | |
| Rural | 10.37073 | 0.416 | Unclassified | 1.517998 | 0.901 |
| Unclassified | 10.37073 | 0.790 | Urban | 2.717691 | 0.894 |
| Both | 5.110404 | 0.633 | Both | 4.400977 | 0.696 |
| Urban | Reference | 1 | Rural | Reference | 1 |
| **Age** | | | **Age** | | |
| 10–19 | 1.350945 | 0.764 | 10–19 | 13.16494 | 0.094 |
| 15–19 | Reference | 1 | 15–19 | Reference | 1 |
| **Sex** | | | **Sex** | | |
| Both | -5.592236 | 0.159 | Both | 7.205115 | 0.295 |
| Female | Reference | 1 | Female | Reference | 1 |
| **Study Design** | | | **Study Design** | | |
| Comparative cross sectional | -11.96162 | 0.107 | Comparative cross sectional | -7.49505 | 0.549 |
| Descriptive cross sectional | Reference | 1 | Descriptive cross sectional | Reference | 1 |
| **Sample size** | | | **Sample size** | | |
| ≥384 | 6.653005 | 0.380 | <384 | -8.788739 | 0.224 |
| <384 | Reference | 1 | ≥384 | Reference | 1 |

(15%) [78]. The prevalence of overweight/obesity was also lower than the study conducted in America (30%), Europe (22%–25%) and Italy(17.9%), Oceania, Australia(23.2%) and New Zealand (34.2%) [79]. This discrepancy could be due to variation in socioeconomic status, the life style consuming energy dense diet and sedentary lifestyles.

Factors associated with overweight/obesity among adolescents were reviewed and meta-analyzed. Females had higher odds of developing overweight/obesity compared to males. Gender differences were not observed in a study conducted in African school learner adolescent [77]. This study is in line with studies conducted in four of twenty five countries in the world [79]. But, it is contrary to studies conducted on seventeen of twenty five countries in the world [79]. This may be explained by biological differences in energy need and body composition between males and females in relation to rate of growth and timing of sexual maturation [25]. Males are also more physically active than females. In developing countries including Ethiopia, girls usually stay at home due to cultural influence not to move from place to place than boys which results in physical inactivity and ultimately lead to overweight and obesity.

Adolescents with low DDS had two times higher odds of being overweight / obese compared to those with higher DDS. Low DDS reflects monotonous dietary intake which can result in overweight/obesity. The reason could be that adolescents have a greater preference to sweet food products which are calorie rich and leading to a positive energy balance.

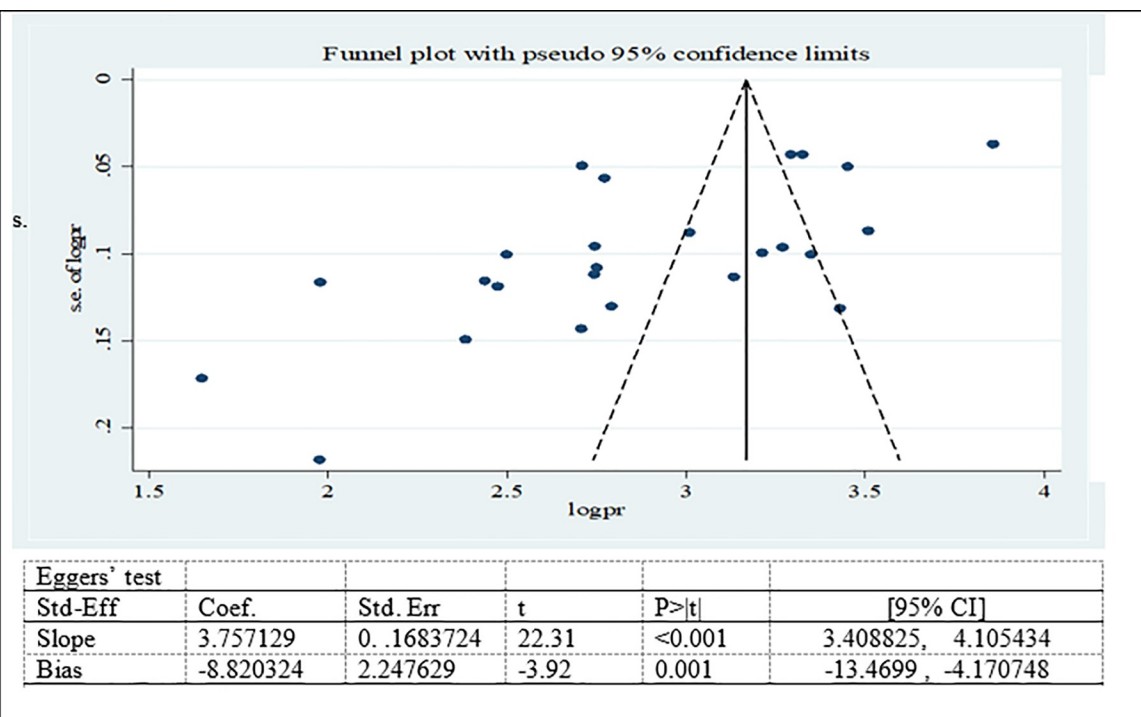

**Fig 8. Funnel plot and Eggers' test to assess publication bias for adolescent stunting in Ethiopia.**

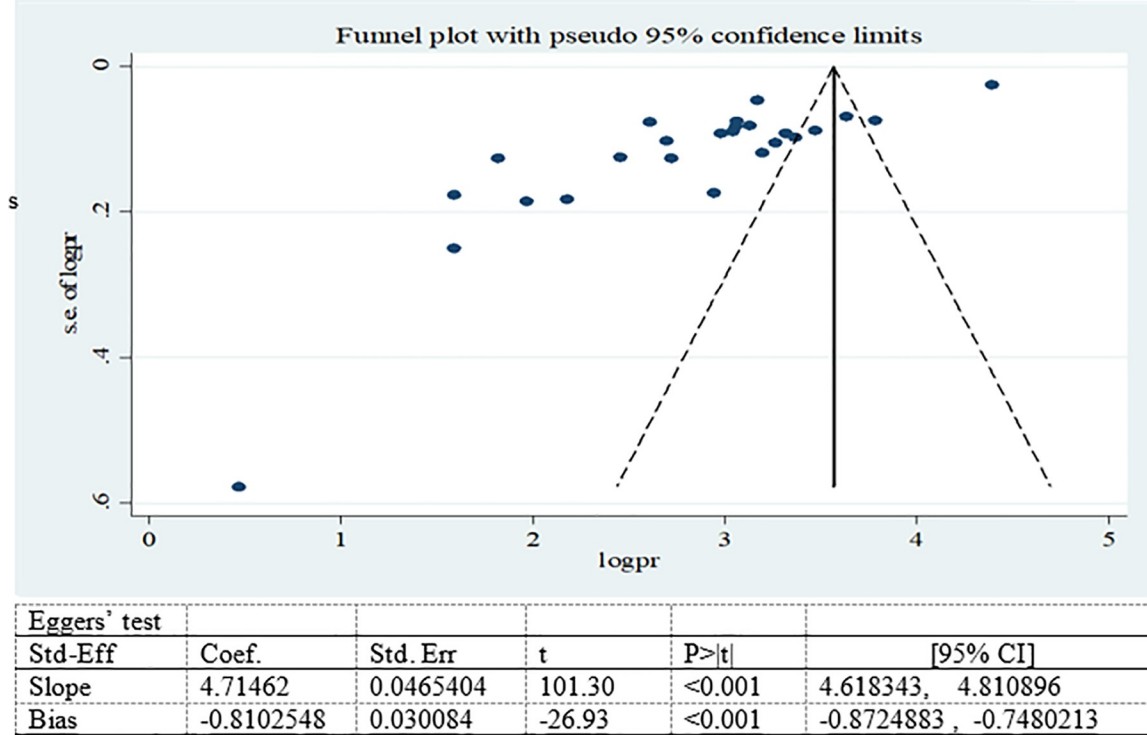

**Fig 9. Funnel plot and Eggers' test to assess publication bias for adolescent thinness in Ethiopia.**

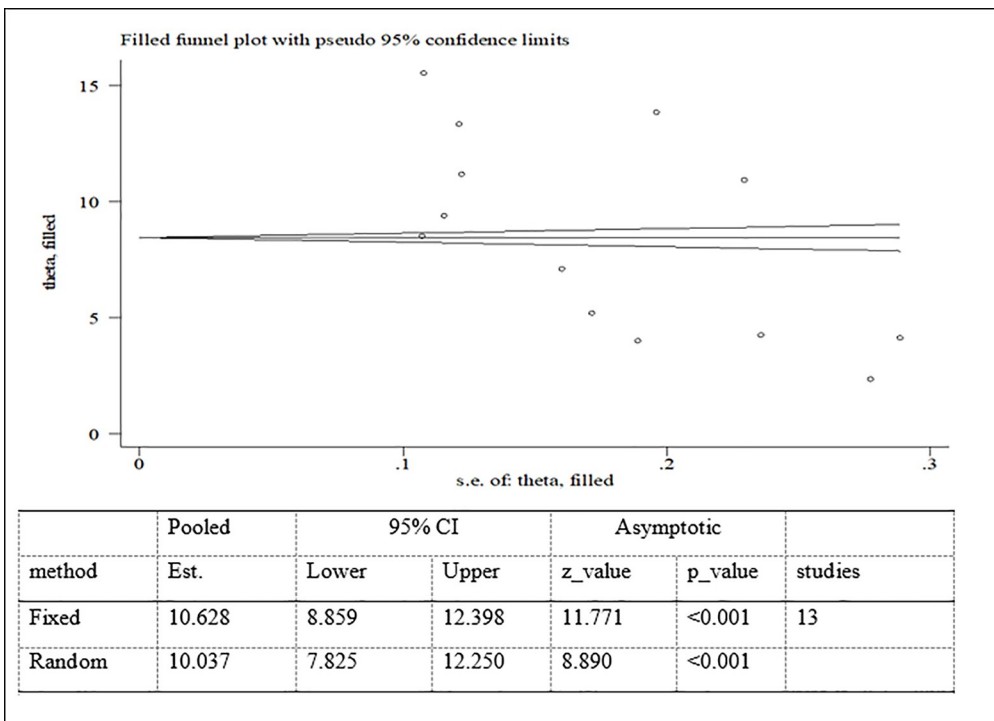

Fig 10. Trim and fill analysis for overweight/obesity among adolescent in Ethiopia.

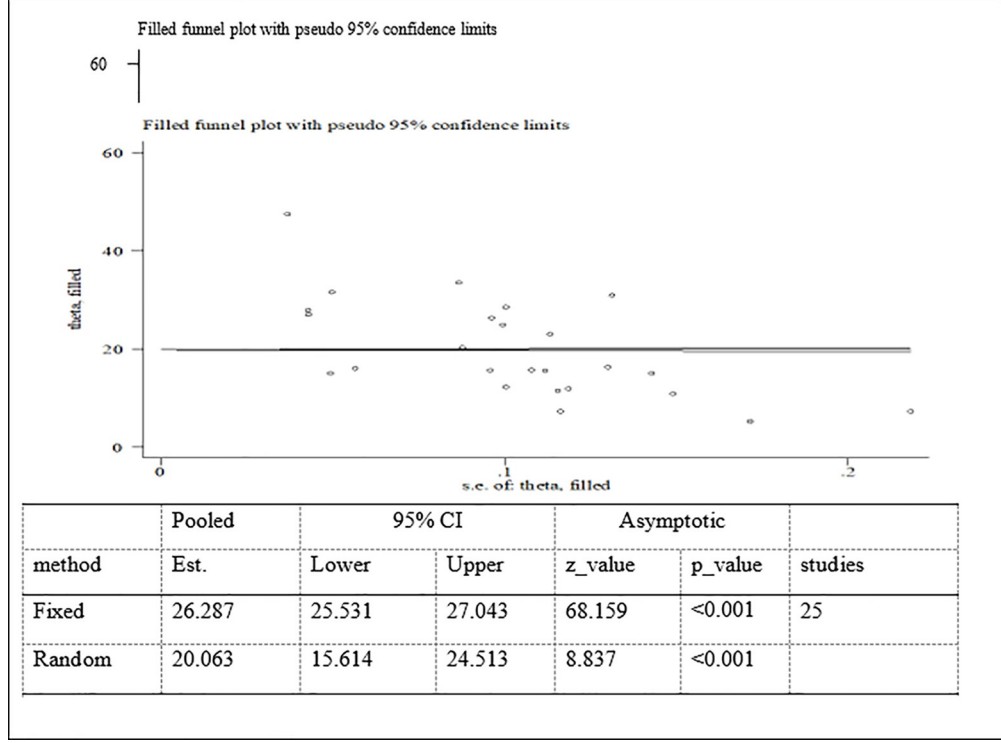

Fig 11. Trim and fill analysis for stunting among adolescent in Ethiopia.

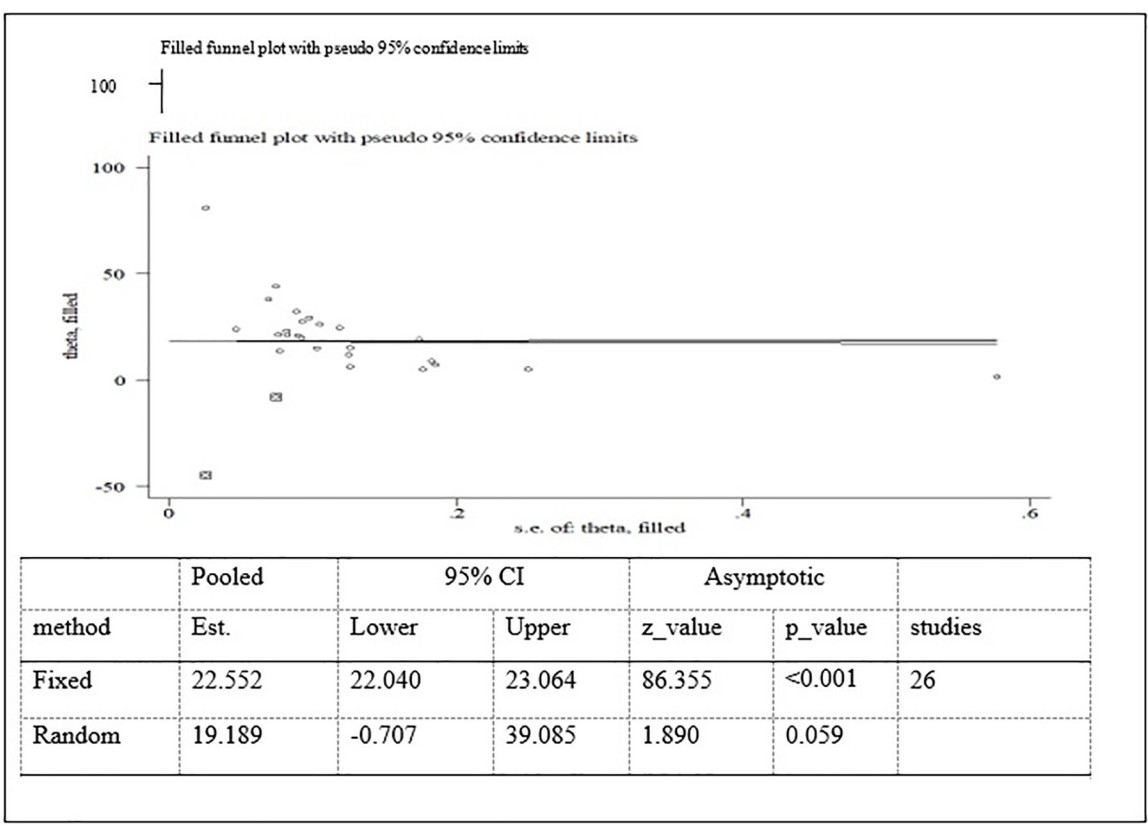

| | Pooled | 95% CI | | Asymptotic | | |
|--------|--------|--------|--------|---------|---------|---------|
| method | Est. | Lower | Upper | z_value | p_value | studies |
| Fixed | 22.552 | 22.040 | 23.064 | 86.355 | <0.001 | 26 |
| Random | 19.189 | -0.707 | 39.085 | 1.890 | 0.059 | |

**Fig 12. Trim and fill analysis for thinness among adolescent in Ethiopia.**

Adolescents with high physical activity had 64% lower odds of being overweight/obese. Physical activities are important for burning fat tissues and increase muscular tissue [77].

The pooled prevalence of adolescent stunting was 20.06% (95% CI: 15.61–24.51). The prevalence of stunting in this study was within the range of the prevalence reported from Latin America and Caribbean countries (6.5–42.7%) [80]. The stunting prevalence of this study was higher than the result reported from studies conducted in 57 low and middle-income countries (10.2%) [81]. The difference could be due to variation in sampling and study period, cultural and dietary practices, access and utilization of health services.

In this analysis, adolescents in the urban areas had 18% lower odds of stunting. This may due to the inequalities in socio-economic status, access to medical services and health information in urban and rural settings. Adolescents from household with protected water source for drinking had 50% lower odds of stunting. Protected source of drinking water is a mechanism to prevent intestinal parasites and other communicable diseases which causes poor nutritional status. Unprotected water source results in repeated infections, depressed immunity increasing the severity and duration of diseases. This finding was similar with study conducted in sub-Sahara African countries [82]. Adolescents in the households with family size of < 5 were 46% less likely to be stunted than those living in the households with family size of ≥5. This might be due to the fact that small family size is usually found in educated parents who are more likely to be aware of dietary diversity and have good dietary consumption practice [48]. There is also enough food among the small household members for adequate consumption.

In this study, the pooled prevalence of adolescent thinness was 21.68% (95% CI: 9.56–33.81).This finding was consistent with EDHS 2005 (32.5%) and 2016 (29%) (78, 80). This

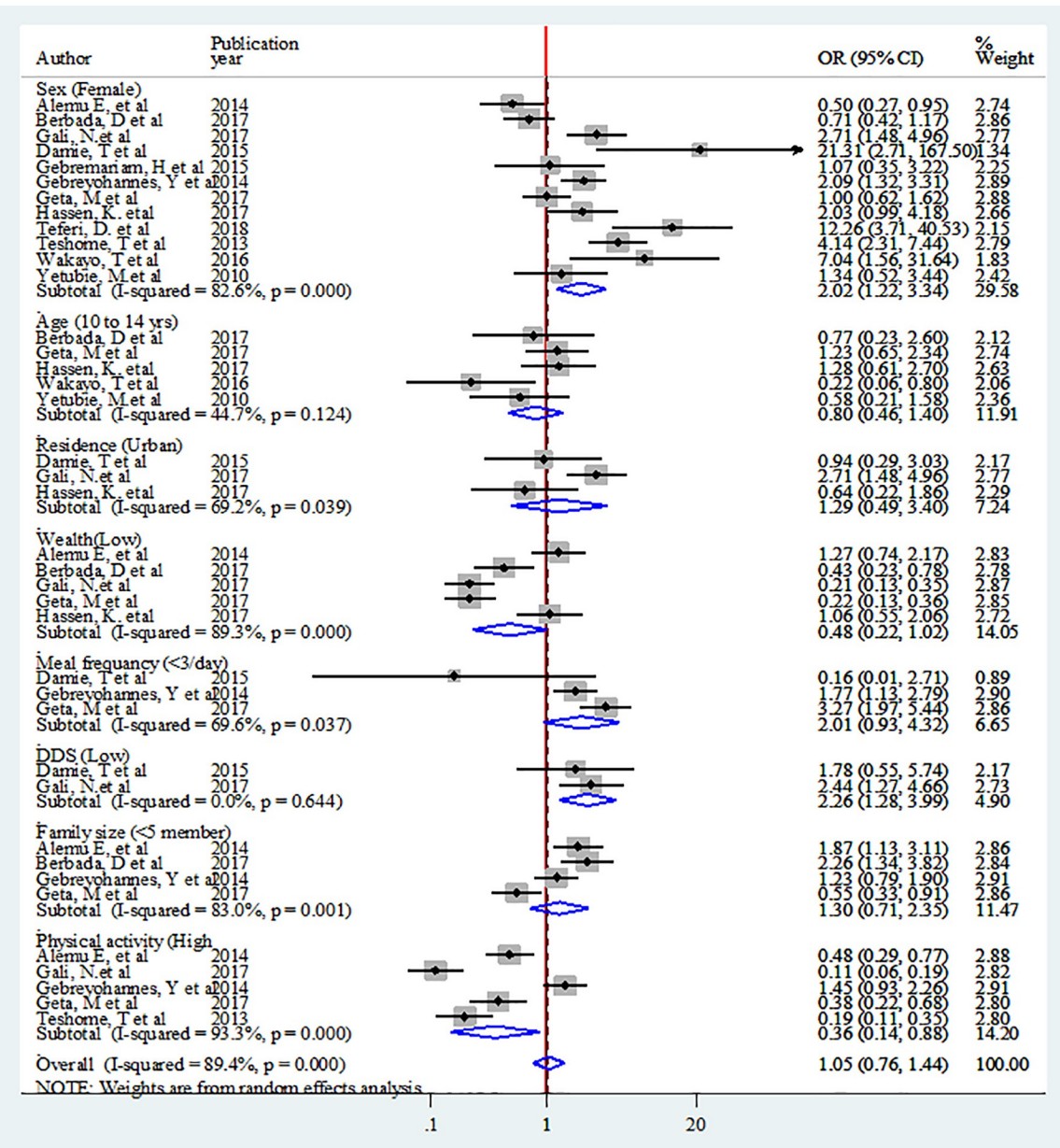

**Fig 13. Pooled Odds Ratios of factors associated with overweight/obesity among adolescent in Ethiopia.**

result was in line with the finding of review on seven African countries(12.6–31.9%) [83] and Latin America and Caribbean countries (3.1–21.6%) [80]. However, prevalence of thinness was higher compared to studies conducted in 57 low and middle-income countries (5.5%) [81]. The possible explanation for this difference could be due variations in socioeconomic status, study period and access to and utilization of health care services.

The effect size of factors associated with adolescent thinness was also estimated. Adolescent in the households with protected water source had 64% less odds of being thin than those who had unprotected water source. This can be explained by the fact that protecting source of drinking water is a mechanism for preventing intestinal parasites and other communicable

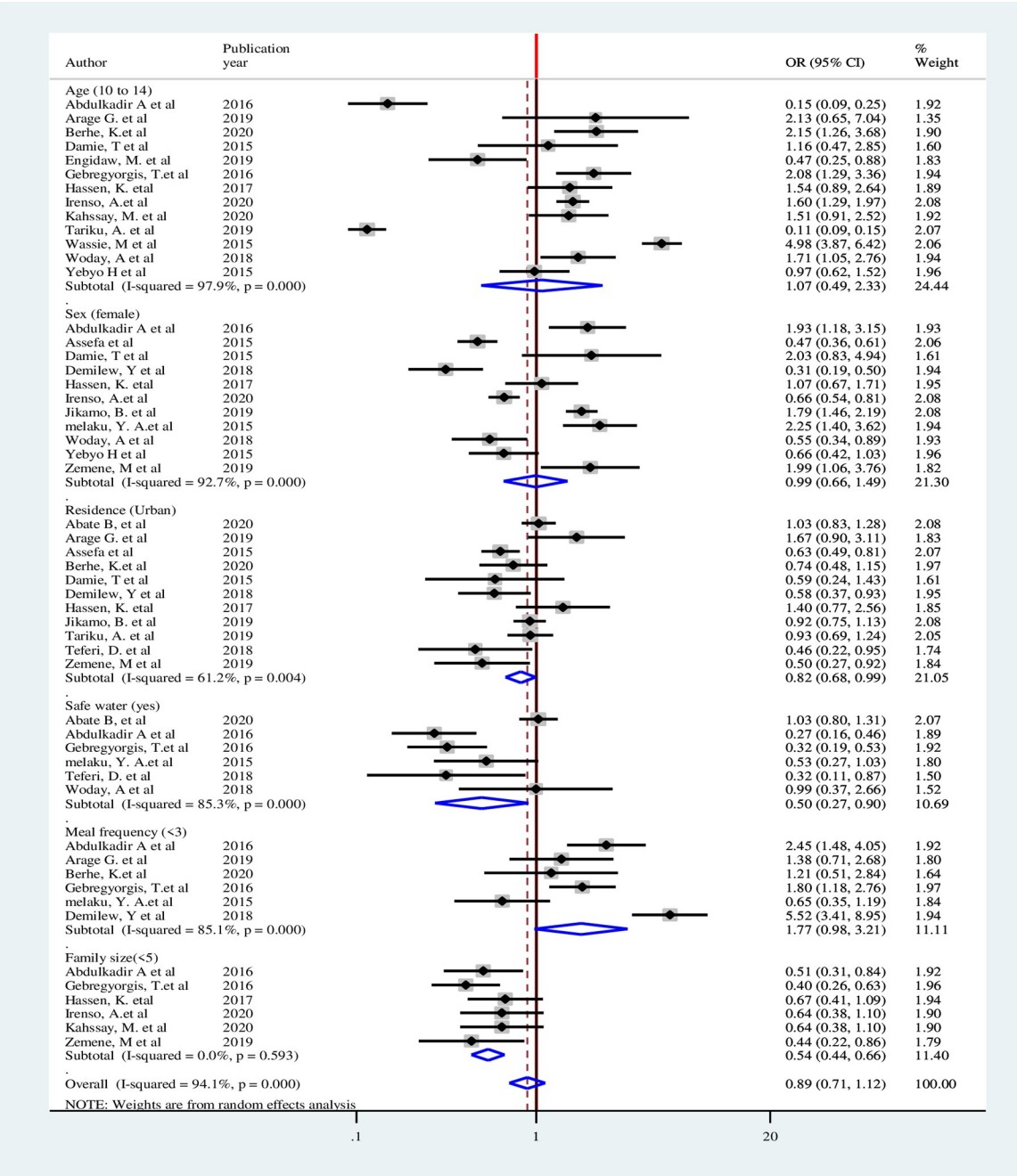

**Fig 14. Pooled Odds Ratio of factors associated with stunting among adolescent in Ethiopia.**

diseases which causes poor nutritional status. Unprotected water source could lead to higher frequencies of infections, depressed immunity and making the severity and duration of diseases [41].

Thinness was identified in early adolescents (10-14years) was 2 times higher than late adolescents. There is faster growth and development in the early age of adolescents (10–14 years) as compared to late adolescents (15–19years) [25].When the requirement nutrient for achieving their maximum need for growth and development is not fulfilled, adolescents would be

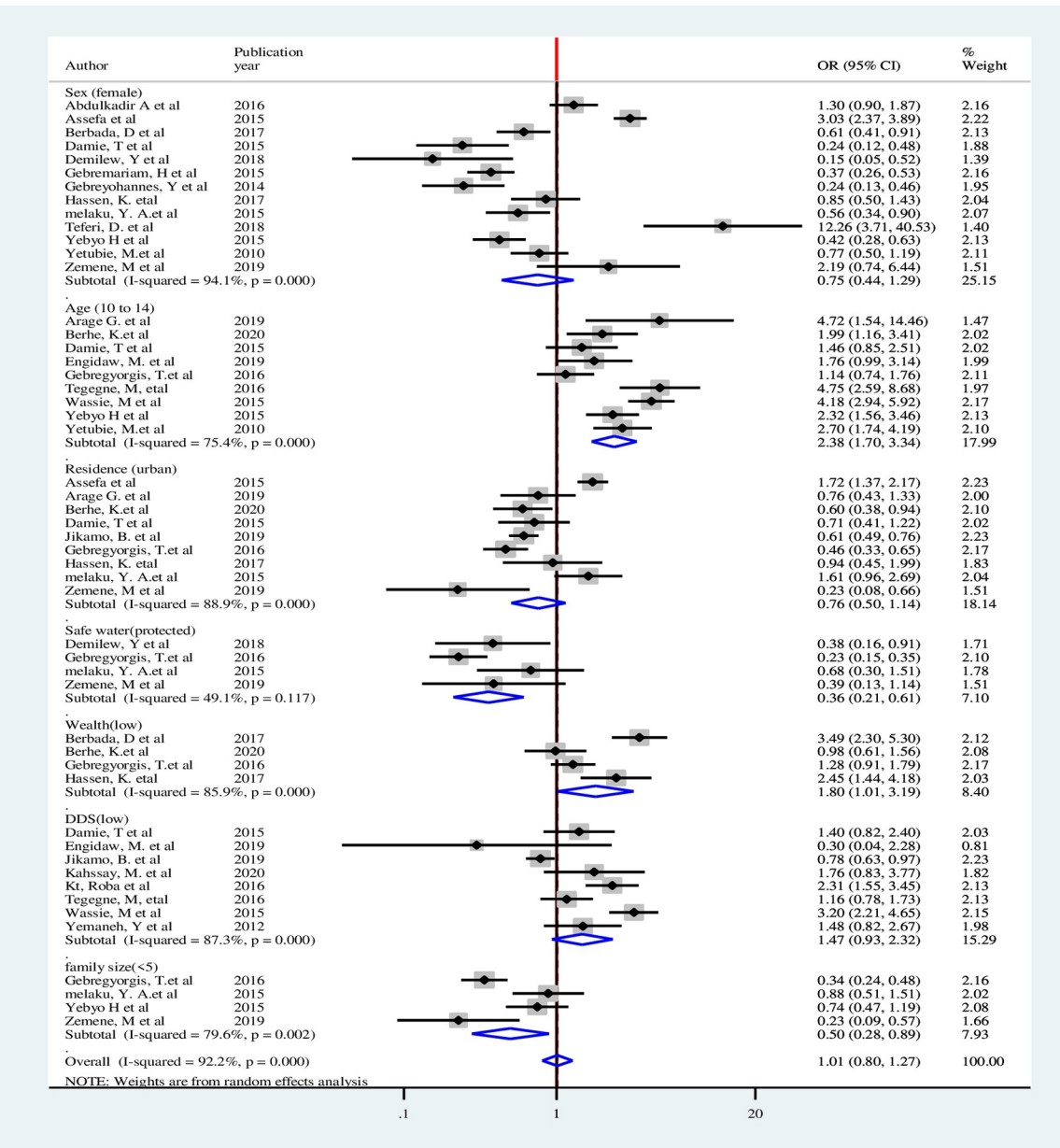

**Fig 15. Pooled Odds Ratio of factors associated with thinness among adolescent in Ethiopia.**

affected by thinness. Adolescent thinness with lower wealth index family income was almost 2 times higher than high wealth index. It might be due to inability to afford food items for consumption and inadequate dietary intake which can result in thinness.

The study has practical implications. Despite methodological differences, nutritional patterns and the availability of recreational facilities, the findings of this analyses clearly indicates that nutrition transition (overweight/ obesity) is becoming a public health problem with the existing undernutrition creating a double burden. This affects educational status, productivity, health status of the community and economic growth of the country at large. Although, there is heterogeneity among the regions, malnutrition in adolescents has taken the status of double burden, which calls for programs and policies to consider this in addressing the nutritional

**Table 6.  Subgroup analysis of factors associated with adolescent malnutrition in Ethiopia.**

| Sub-group | | Variables | Number of studies | OR (95% CI) | Heterogeneity | |
|---|---|---|---|---|---|---|
| | | | | | I²% | P |
| **Overweight/Obesity** | | | | | | |
| **Sex** | | **Overall** | **12** | **2.02(1.22–3.34)** | **82.6** | **<0.001** |
| | Region | Adis Ababa | 2 | 1.04(0.26, 4.22) | 92.0 | <0.001 |
| | | SNNP | 4 | 2.22(0.78, 6.32) | 91.3 | <0.001 |
| | | Oromo | 5 | 2.82(1.51, 5.27) | 50.2 | 0.091 |
| | | Tigray | 1 | 1.07(0.35, 3.22) | - | - |
| | Study design | Cross-sectional | 9 | 2.29(1.09, 4.80) | 85.4 | <0.001 |
| | | Other | 2 | 1.45(0.70, 2.99) | 78.6 | <0.001 |
| | | Overall | 11 | 1.98(1.15, 3,42) | 83.1 | <0.001 |
| | Sample size | > = 384 | 8 | 1.72(0.92, 3.21) | 84.6 | <0.001 |
| | | <384 | 3 | 4.46(0.61, 32.35) | 85.5 | 0.001 |
| **Physical activity** | | **Overall** | **5** | **0.36(0.14–0.88)** | **93.3** | **<0.001** |
| | Region | AA | 2 | 0.83(0.28, 2.48) | 90.9 | 0.001 |
| | | Oromo | 1 | 0.11(0.06, 0.19) | - | - |
| | | SNNP | 2 | 027(0.14, 0.54) | 63.7 | 0.097 |
| | Study design | Cross-sectional | 3 | 0.22(0.09, 0.52) | 87.6 | <0.001 |
| | | Other | 2 | 0.75(0.20, 2.78) | 92.2 | <0.001 |
| | Sample size | > = 384 | 4 | 0.35(0.11, 1.10) | 95.0 | <0.001 |
| | | <384 | 1 | 0.38(0.22, 0.68) | - | - |
| | | Overall | 5 | 0.36(0.14, 0.88) | 93.3 | <0.001 |
| **Stunting** | | | | | | |
| **Residence** | | **Overall** | **11** | **0.82(0.68–0.99)** | **61.2** | **0.004** |
| | Region | Other | 3 | 0.79(0.52, 1.18) | 63.2 | 0.066 |
| | | Amhara | 4 | 0.82(0.52, 1.27) | 70.9 | 0.016 |
| | | Oromo | 4 | 0.83(0.60, 1.15) | 66.8 | 0.029 |
| | Study Design | Cross-sectional | 10 | 0.82(0.68, 0.99) | 61.2 | 0.004 |
| | | Overall | 10 | 0.82(0.68, 0.99) | 61.2 | 0.004 |
| | Sample size | > = 384 | 8 | 0.82(0.68, 0.99) | 60.4 | 0.014 |
| | | <384 | 4 | 0.80(0.36, 1.81) | 75.4 | 0.017 |
| | | Overall | 12 | 0.82(0.68, 0.99) | 61.2 | 0.004 |
| **Safe water** | | **Overall** | **6** | **0.50(0.27–0.90)** | **85.3** | **<0.001** |
| | Region | Others | 4 | 0.55(0.24,1.27) | 86.9 | <0.001 |
| | | Tigray | 3 | 0.39(0.24,0.90) | 26.4 | 0.244 |
| | Study design | Descriptive Cross-sectional | 5 | 0.57(0.31, 1.05) | 81.3 | <0.001 |
| | | Comparative Cross-sectional | 1 | 0.27(0.16, 0.46) | - | - |
| | | Overall | 6 | 0.49(0.27, 0.90) | 85.3 | <0.001 |
| | Sample size | ≥384 | 5 | 0.49(0.24, 1.00) | 88.1 | <0.001 |
| | | <384 | 1 | 0.53(0.27, 1.03) | - | |
| | | Overall | 6 | 0.50(0.27, 0.90) | 85.3 | <0.001 |
| **Thinness** | | | | | | |
| **Age** | | **Overall** | **11** | **2.38(1.70–3.34)** | **75.4** | **<0.001** |

(*Continued*)

**Table 6.** (Continued)

| Sub-group | | Variables | Number of studies | OR (95% CI) | Heterogeneity | |
|---|---|---|---|---|---|---|
| | | | | | I²% | P |
| | Region | Amhara | 2 | 4.22(3.02,5.89) | 0.0 | 0.0.836 |
| | | Tigray | 3 | 1.74(1.11,2.72) | 66.3 | 0.051 |
| | | Oromo | 3 | 2.63(1.41,4.87) | 75.9 | 0.016 |
| | | Somalia | 2 | 1.76(0.99,3.14) | - | - |
| | | Overall | 10 | 2.38(1.70,3.34) | 75.4 | <0.001 |
| | Study design | Cross-sectional | 10 | 2.38(1.70, 3.34) | 75.4 | <0.001 |
| | | Overall | 10 | 2.38(1.70, 3.34) | 75.4 | <0.001 |
| | Sample size | > = 384 | 7 | 2.42(1.66, 3.53) | 78.4 | <0.001 |
| | | <384 | 2 | 2.36(0.76, 7.32) | 70.8 | 0.064 |
| | | Overall | 9 | 2.38(1.69, 3.34) | 75.4 | <0.001 |
| **Wealth (low)** | | **Overall** | **4** | **1.80(1.01–3.19)** | **85.9** | **<0.001** |
| | Region | Other | 2 | 3.05(2.18, 4.25) | 3.7 | 0.308 |
| | | Tigray | 2 | 1.17(0.89, 1.53) | 0.0 | 0.368 |
| | Study design | Cross-sectional | 4 | 1.80(1.01, 3.19) | 85.9 | <0.001 |
| | | Overall | 4 | 1.80(1.01, 3.19) | 85.9 | <0.001 |
| | Sample size | > = 384 | 4 | 1.80(1.01, 3.19) | 85.9 | <0.001 |
| | | Overall | 4 | 1.80(1.01, 3.19) | 85.9 | <0.001 |
| **Family size** | | **Overall** | **4** | **0.50(0.28–0.89)** | **79.6** | **0.002** |
| | Region | Tigray | 3 | 0.59(0.31, 1.11) | 83.2 | 0.003 |
| | | Amhara | 1 | 0.23(0.09, 0.57) | - | - |
| | Study design | Cross-sectional | 4 | 0.50(0.28, 0.89) | 79.6 | 0.002 |
| | | Overall | 4 | 0.50(0.28, 0.89) | 79.6 | 0.002 |
| | Sample size | > = 384 | 2 | 0.49(0.23, 1.07) | 85.9 | 0.008 |
| | | <384 | 2 | 0.47(0.13, 1.78) | 84.0 | 0.012 |
| | | Overall | | 0.50(0.28, 0.89) | 79.6 | 0.002 |

status of this segment of population. For researchers there is a need for surveillance of the problem to track the trend and risk factors for nutritional status among adolescents in Ethiopia.

This study did a comprehensive review of studies which assessed nutritional status (overweight/obesity, stunting and thinness) and the effect of gender on nutritional status of adolescents. The use of multiple reputable databases, reproducible and pretested extraction formats and inclusion of studies from different regions of the country can are some of the strengths of the study. However, we acknowledge limitations inclosing use of mostly cross-sectional studies could affect the temporal relationship between the assessed determinant factors and outcome variables. Most of the studies were institution/school based may not represent spume pout of school adolescents although they are few as over 90% adolescents are in school. The numbers of studies for estimation of the effect size of associated factors were small which could affect the generalization of the findings. Study participants were not proportional in sex (female participants were higher in number).The study was restricted to articles only published in English language. Heterogeneity was high, despite the use of random effects models to accommodate this variability. Even if trim and fill analysis was done, the funnel plots reported high publication bias. Therefore, findings of this review and meta-analysis should be interpreted with caution in the context of the inherent limitations of both the original studies and the present review and meta-analysis.

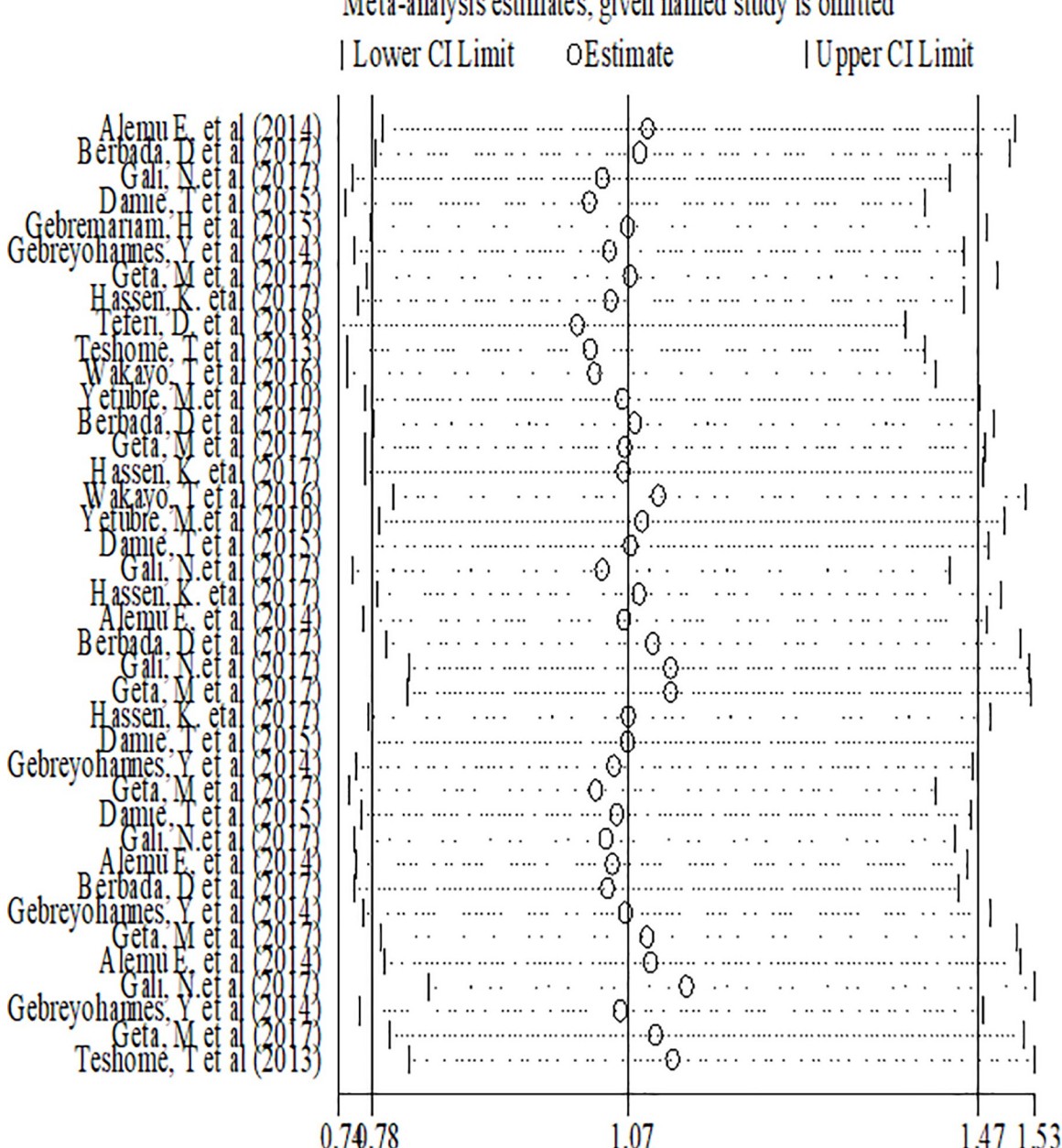

**Fig 16. Sensitivity analysis of factors associated with overweight/obesity in Ethiopia.**

## Conclusion

Adolescent nutritional status remains one of the most important public health problems in Ethiopia. The pooled estimate of overweight/obesity showed high increase, with the existing high burden of stunting and thinness. Female sex, DDS and physical activity were factors significantly associated with adolescent overweight/obesity. Urban residence, family size < 5 family and protected drinking water source were predictors of adolescent stunting. Being early adolescent (10–14 years), wealth index, protected water source and family size < 5 members were significantly associated with adolescent thinness.

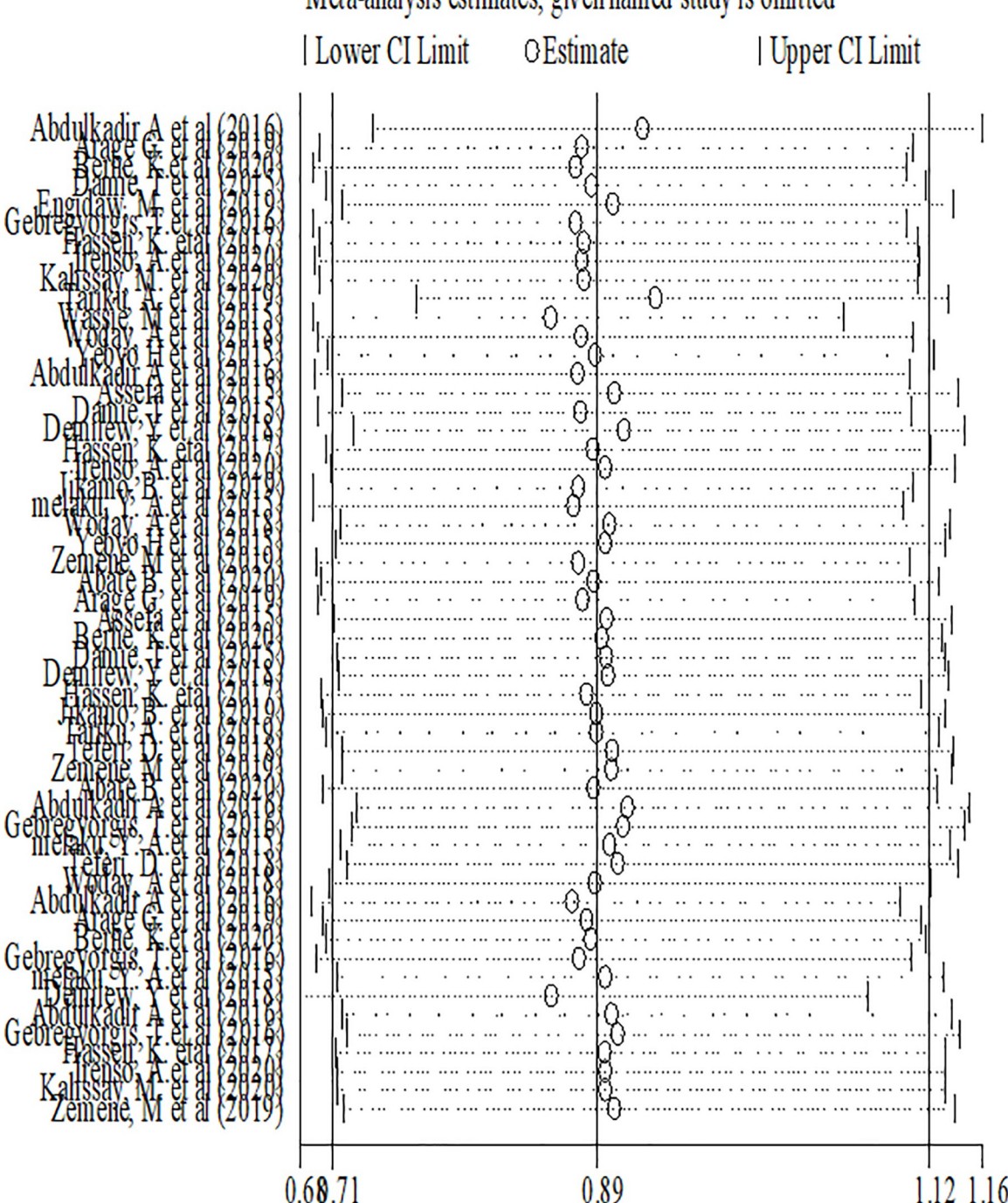

**Fig 17. Sensitivity analysis of factors associated with stunting in Ethiopia.**

The results imply the need for giving more emphasis on design and implementation of preventive policies to reduce stunting and thinness and the high prevalence of overweight/obesity among adolescent in Ethiopia. It is imperative to design interventions that address the emerging dual burden of malnutrition through providing comprehensive and routine nutritional assessment and counseling services at health facility, school and community levels.

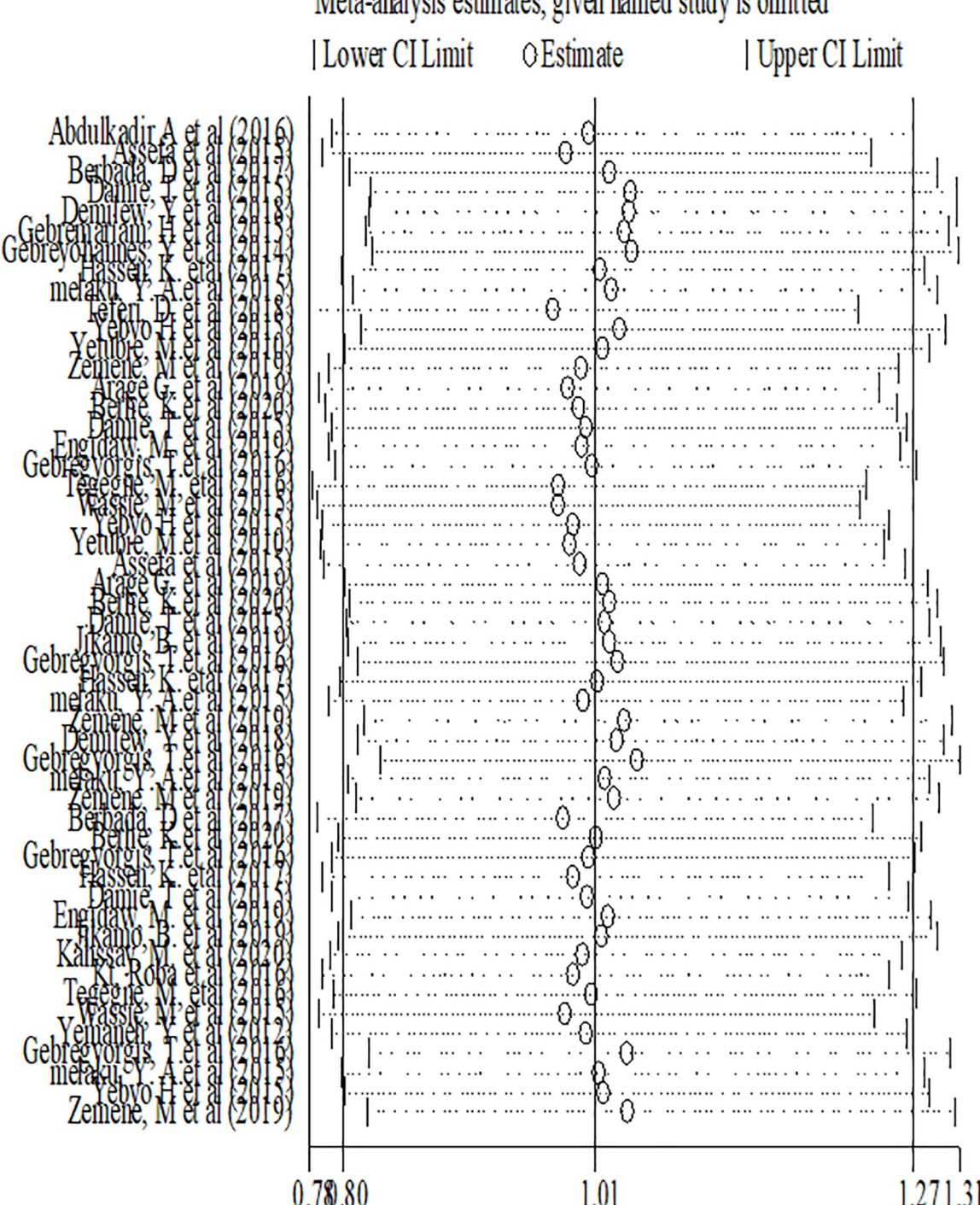

**Fig 18. Sensitivity analysis of factors associated with thinness in Ethiopia.**

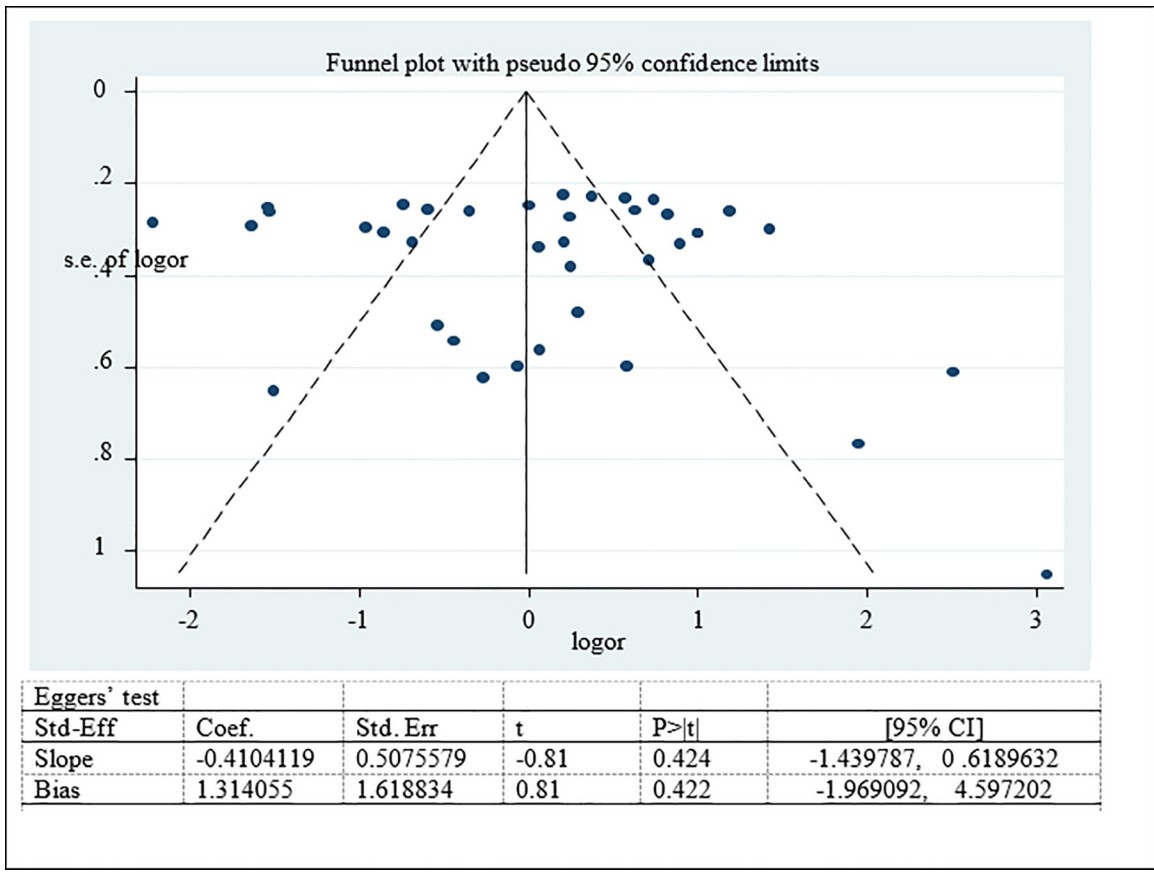

**Fig 19. Funnel plot and Eggers' test to assess publication bias for factor associated with adolescent overweight/obesity in Ethiopia.**

**Table 7. Meta-regression to identify heterogeneity among factors associated with adolescent malnutrition in Ethiopia.**

| Variables | Coefficients | P |
|---|---|---|
| **Overweight/obesity** | | |
| Publication year | 0.0159467 | 0.871 |
| Sample size | 0.0004868 | 0.547 |
| Age(10–14) | -0.6196052 | 0.819 |
| DDs(low) | 0.6712044 | 0.843 |
| Family size (<5) | 0.0391229 | 0.989 |
| Physical activity(high) | -0.9172185 | 0.735 |
| Sex(female) | 3.131227 | 0.197 |
| Wealth(low) | -0.8022954 | 0.767 |
| Residence(urban) | Reference | 1 |
| **Stunting** | | |
| Publication year | 0.0139177 | 0.827 |
| Sample size | -0.0000627 | 0.711 |
| Age(10–14) | 1.010321 | 0.042 |
| Family size (<5) | -0.0254711 | 0.964 |

*(Continued)*

**Table 7.** (Continued)

| Variables | Coefficients | P |
|---|---|---|
| Residence (urban) | 0.2860214 | 0.567 |
| Sex (female) | 0.6528639 | 0.194 |
| Protected water | Reference | 1 |
| **Thinness** | | |
| Publication year | -0.0331687 | 0.561 |
| Sample size | 0.0002983 | 0.294 |
| Age(10–14) | 1.095686 | 0.050 |
| Residence(urban) | 0.3036364 | 0.783 |
| Protected water | -0.1338743 | 0.918 |
| Sex(female) | 1.155758 | 0.274 |
| Wealth(low) | 1.498451 | 0.250 |
| Family size(<5) | Reference | 1 |

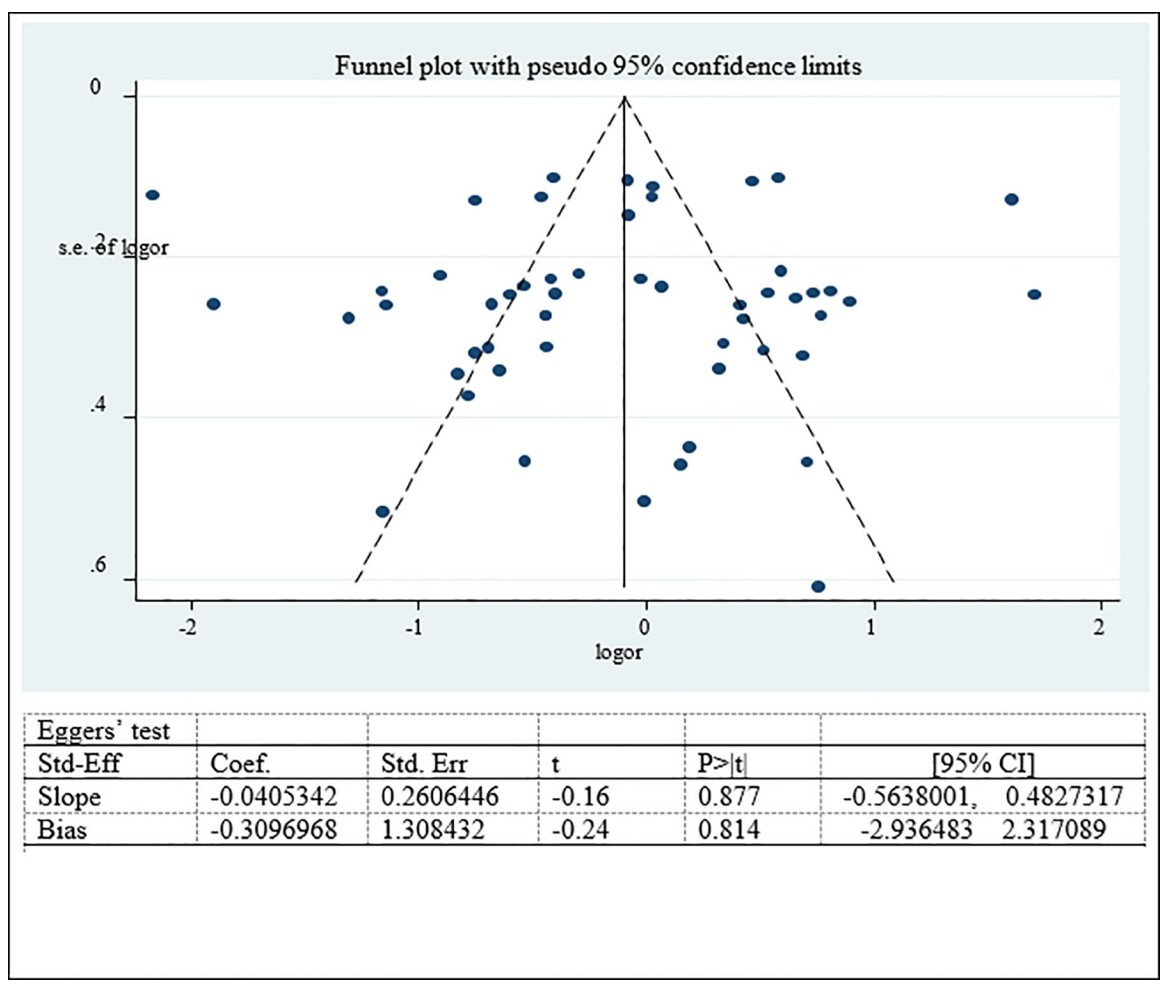

**Fig 20. Funnel plot and Eggers' test to assess publication bias for factor associated with adolescent stunting in Ethiopia.**

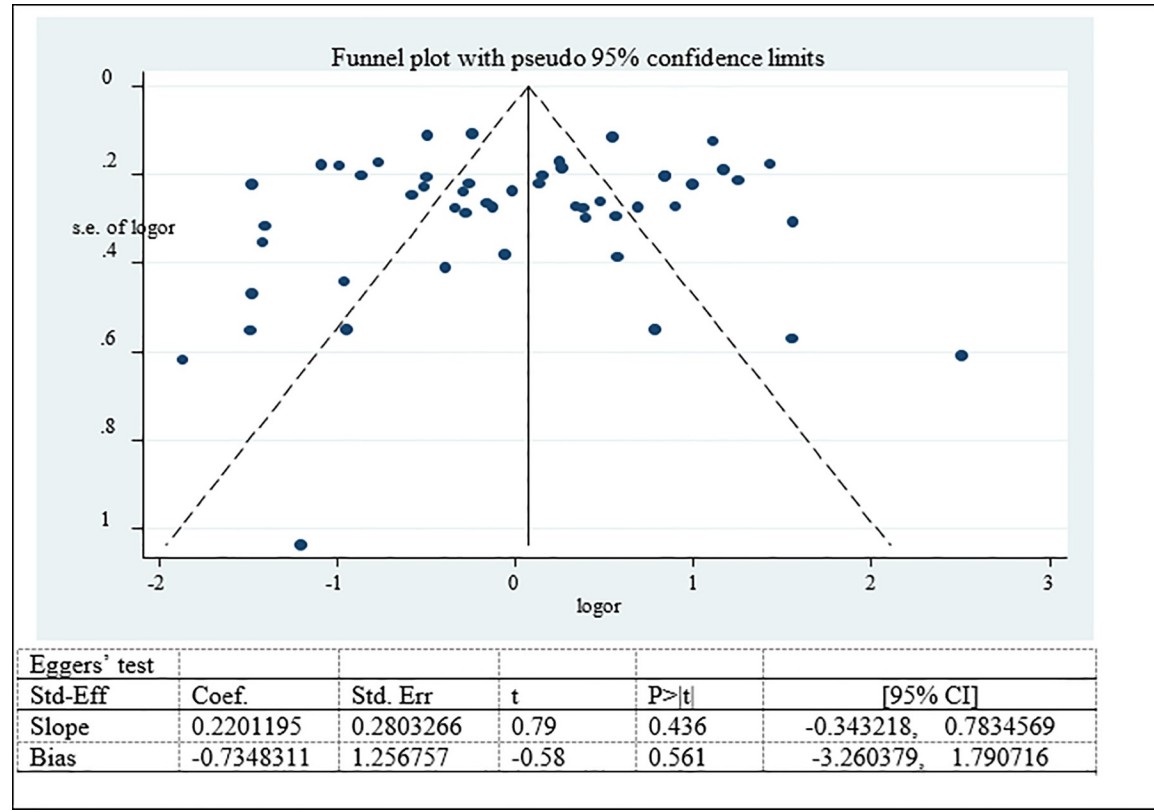

**Fig 21. Funnel plot and Eggers' test to assess publication bias for factor associated with adolescent thinness in Ethiopia.**

## Supporting information

**S1 Checklist. PRISMA 2009 checklist.**
(DOC)

**S1 Table. JBI critical appraisal checklist for studies included in systematic review and meta-analysis of adolescent nutritional status in Ethiopia.**
(DOCX)

**S1 File.**
(DOC)

## Acknowledgments

We would like to extend our gratitude to the authors of the primary papers, study participants and the data collectors.

## Author Contributions

**Conceptualization:** Aragaw Gezaw, Wolde Melese, Bekalu Getachew, Tefera Belachew.

**Data curation:** Aragaw Gezaw, Wolde Melese, Bekalu Getachew, Tefera Belachew.

**Formal analysis:** Aragaw Gezaw, Wolde Melese, Bekalu Getachew, Tefera Belachew.

**Funding acquisition:** Aragaw Gezaw, Wolde Melese, Bekalu Getachew, Tefera Belachew.

**Investigation:** Aragaw Gezaw, Wolde Melese, Bekalu Getachew, Tefera Belachew.

**Methodology:** Aragaw Gezaw, Wolde Melese, Bekalu Getachew, Tefera Belachew.

**Project administration:** Aragaw Gezaw, Bekalu Getachew, Tefera Belachew.

**Resources:** Aragaw Gezaw, Tefera Belachew.

**Software:** Aragaw Gezaw, Bekalu Getachew, Tefera Belachew.

**Supervision:** Aragaw Gezaw, Wolde Melese, Tefera Belachew.

**Validation:** Aragaw Gezaw, Wolde Melese, Bekalu Getachew, Tefera Belachew.

**Visualization:** Aragaw Gezaw, Bekalu Getachew, Tefera Belachew.

**Writing – original draft:** Aragaw Gezaw, Wolde Melese, Bekalu Getachew, Tefera Belachew.

**Writing – review & editing:** Aragaw Gezaw, Wolde Melese, Bekalu Getachew, Tefera Belachew.

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
