## [Decision Letter · Decision Letter 0]

22 Sep 2021

PONE-D-21-00351Double burden of malnutrition and Associated Factors among Adolescent in Ethiopia: A Systematic Review and Meta AnalysisPLOS ONE

Dear Dr. Gezaw,

Thank you for submitting your manuscript to PLOS ONE. After careful consideration, we feel that it has merit but does not fully meet PLOS ONE’s publication criteria as it currently stands. Therefore, we invite you to submit a revised version of the manuscript that addresses the points raised during the review process.

The manuscript has been evaluated by two reviewers, and their comments are available below.

The reviewers have raised a number of concerns regarding the manuscript’s clarity and organization. They specifically request discussion of the study’s limitations throughout the article, as well as some statistical suggestions.

Could you please carefully revise the manuscript to address all comments raised?

We look forward to receiving your revised manuscript.

Kind regards,

Avanti Dey, PhD

Staff Editor

PLOS ONE

Journal Requirements:

2. Thank you for stating the following in the Acknowledgments Section of your manuscript: "This research was funded by Amhara Health Bureau. We would like to extend our gratitude to Amhara Health Bureau for funding this study. We are also grateful to the authors of the primary papers, study participants and the data collectors."

Please remove any funding-related text from the manuscript and let us know how you would like to update your Funding Statement. Currently, your Funding Statement reads as follows: "This research was funded by Amhara Health Bureau. We would like to extend our gratitude to Amhara Health Bureau for funding this study."

5. Please upload a new copy of Figures 14 and 15 as the detail is not clear. Please follow the link for more information: " ext-link-type="uri" xlink:type="simple">https://blogs.plos.org/plos/2019/06/looking-good-tips-for-creating-your-plos-figures-graphics/"
" ext-link-type="uri" xlink:type="simple">https://blogs.plos.org/plos/2019/06/looking-good-tips-for-creating-your-plos-figures-graphics/".

6. Please upload a copy of Figures 19, 20 and 21 to which you refer in your text on page 8. If the figure is no longer to be included as part of the submission please remove all reference to it within the text.

7. We note you have included a table to which you do not refer in the text of your manuscript. Please ensure that you refer to Table 8 in your text; if accepted, production will need this reference to link the reader to the Table.

8.Please upload a copy of Supporting Information Table 4 which you refer to in your text on page 12.

9. We note that this manuscript is a systematic review or meta-analysis; our author guidelines therefore require that you use PRISMA guidance to help improve reporting quality of this type of study. Please upload copies of the completed PRISMA checklist as Supporting Information with a file name “PRISMA checklist”.

Reviewers' comments:

Reviewer's Responses to Questions

**Comments to the Author**

1. Is the manuscript technically sound, and do the data support the conclusions?

Reviewer #1: Partly

Reviewer #2: Yes

2. Has the statistical analysis been performed appropriately and rigorously? 

Reviewer #1: Yes

Reviewer #2: Yes

3. Have the authors made all data underlying the findings in their manuscript fully available?

Reviewer #1: No

Reviewer #2: Yes

4. Is the manuscript presented in an intelligible fashion and written in standard English?

Reviewer #1: Yes

Reviewer #2: No

5. Review Comments to the Author

Reviewer #1: - There were not the tables 1 to 4 in the file and I couldn’t see the list of studies and their score of critical appraisals.

- One of the most important of restrictions of this study is that it was restricted to only ‘English language’ which can lead to language bias and unreal estimation. It is recommended to include all published studies of Ethiopia adolescents regardless of the language of the articles

- Other weakness of study is that it included all studies without time restriction, while the malnutrition is a time depended issue and in order to get a true picture of the current state of society, it is better to apply a time restriction or do a cumulative meta-analysis to observe the changes in the index over time.

- It is recommended to use the random-effects model even if the heterogeneity is low, so it is recommended to use the random model for overweight/ obesity in this study as well.

- Have studies whose reports are not based on Z- Score been deleted?

Reviewer #2: Dear Editor,

Thank you, for giving me the opportunity to review this manuscript, which sets out to assess the prevalence and the associated factors of the double burden of malnutrition in Ethiopia, using meta-analysis and systematic review approach. The paper is interesting and largely well presented. However, there are shortfalls that need to be addressed before it can be publishable. Please, find details of my review below:

Abstract:

• Indicate the direction of effects of the independent variables on the dependent variable. Or better still, present the results in the context of ORs.

Introduction:

• The referencing throughout this section needs to be reworked on. All substantial claims should be referenced. E.g., reference is needed in the first part of the second paragraph etc. etc.

• What is the research question(s) the study addresses? SR/MA usually starts by identifying research questions to be addressed

Methods:

• There are so many sub-headings in this section and some of the text overlaps. A critical review of the entire section may be required

• The “data collection process” could be changed to “data extraction process”

• Including the initials of researchers who undertake various assignments is unnecessary…this should be captured under the ‘authors contribution’ section.

• “Operational definition” could be changed to “outcome variables”

• The sub-heading “summary measures” appears misleading as the text that follows does not reflect the heading. Consider revising.

• “Since the data are dichotomous, effect size was calculated for prevalence” What does this mean. Is it suggesting that the only reason for computing the effect size is the dichotomous nature of the data? Clarify.

• No ethical approval was obtained before the conduct of the study, therefore indicating that ethical clearance was given by School of Public Health is misleading. The school granting you access to full articles does not constitute ethics approval. Consider revising the statement.

Results

• The study selection processes may be suitable in the method section than the results section. Consider incorporating part of the text in the method section

• This section is unnecessarily long and difficult to follow. This is probably because of the authors attempt to present all results. I will suggest they present only the striking findings

• Similarly, the authors attempt to explain how the analysis was done to obtain the results has also made the section difficult to read. This should be done in the method section. The results section is meant for the presentation of the findings obtained in the study

• The presentation of the results should be as concise as possible

• The several sub-sections contained in the results section is unnecessary. As I indicated earlier, not all results need to be presented. Consider revising

Discussion

• The presentation of the discussion section is a bit disjointed, thereby making it difficult read. A lot more work is needed to make it flow coherently. The section needs a complete overhaul

• The first paragraph needs to be revised. This paragraph should summarise the main findings but not the literature as has been done

• Restating the results (e.g., CI, ORs etc) is unnecessary. This style has made the section to look more like the results section…take out all CIORs etc and minimise the use of statistics in this section

• Also avoid causal words such as “affecting”

• Further, avoid comparing the prevalence obtained in your study to others in the literature. There are no bases for doing that. Discuss your findings and relate the same to the existing literature

Tables:

• The tables are too many. Not all of them are necessary. The authors should include only those they consider critical

• The Y-axis of tables 13-15 are a bit messy. If the authors want to include the same, they should consider enhancing their readability.

Verdict: The manuscript has the potential to be a good piece of work. However, a lot more effort is needed to bring it to that state. The authors should therefore revise the manuscript critically, and ensure that unnecessarily details etc etc. are minimised. There are also a lot of grammatically issues that need the attention of the authors.

6. PLOS authors have the option to publish the peer review history of their article (what does this mean?). If published, this will include your full peer review and any attached files.

Reviewer #1: **Yes: **Maryam Zamanian

Reviewer #2: **Yes: **Dr Dickson A Amugsi

---

## [Author Response · Author response to Decision Letter 0]

23 Feb 2022

1. We have correctly cited all the tables and figures. Each figure caption appear directly after the paragraph in which they are first cited

2. Language restriction is due to interpretation purpose and almost all studies in Ethiopia published in English language. So language bias is less likely.

3. The protocol of this study was assuming that malnutrition is time dependent and plan to do cumulative meta-analysis, but all the studies done in Ethiopia were recent and we conceder no change among those studies over time.

---

## [Decision Letter · Decision Letter 1]

7 Nov 2022

PONE-D-21-00351R1Double burden of malnutrition and associated factors among adolescent in Ethiopia: A systematic review and meta-analysisPLOS ONE

Dear Dr. Gezaw,

Thank you for submitting your manuscript to PLOS ONE. After careful consideration, we feel that it has merit but does not fully meet PLOS ONE’s publication criteria as it currently stands. Therefore, we invite you to submit a revised version of the manuscript that addresses the points raised during the review process.

Please be sure to address the important recommendations provided by the two expert reviewers. Reviewer 3 especially has provided detailed guidance that will improve the quality and the impact of the manuscript.  As reviewer 4 notes, please also carefully review the grammar and orthography of the manuscript.

We look forward to receiving your revised manuscript.

Kind regards,

Solveig A. Cunningham, Ph.D.

Academic Editor

PLOS ONE

Journal Requirements:

Additional Editor Comments:

Dear authors,

Thank you for the re-submission of your paper after having addressed reviewer comments. The paper has been additionally reviewed by two expert reviewers who have provided excellent and detailed recommendations for further improving the quality and impact of the paper. Please address the comments from the two reviewers.

Reviewers' comments:

Reviewer's Responses to Questions

**Comments to the Author**

1. If the authors have adequately addressed your comments raised in a previous round of review and you feel that this manuscript is now acceptable for publication, you may indicate that here to bypass the “Comments to the Author” section, enter your conflict of interest statement in the “Confidential to Editor” section, and submit your "Accept" recommendation.

Reviewer #3: (No Response)

Reviewer #4: All comments have been addressed

2. Is the manuscript technically sound, and do the data support the conclusions?

Reviewer #3: Partly

Reviewer #4: Yes

3. Has the statistical analysis been performed appropriately and rigorously? 

Reviewer #3: I Don't Know

Reviewer #4: Yes

4. Have the authors made all data underlying the findings in their manuscript fully available?

Reviewer #3: (No Response)

Reviewer #4: Yes

5. Is the manuscript presented in an intelligible fashion and written in standard English?

Reviewer #3: Yes

Reviewer #4: Yes

6. Review Comments to the Author

Reviewer #3: Thank you for the opportunity to review this manuscript on the important topic of adolescent nutrition in Ethiopia. The authors conducted a systematic review and meta-analysis to estimate the pooled prevalence of 3 forms of malnutrition (overweight/obesity, stunting, thinness) and the association between select factors and each of these forms of malnutrition. The topic of the paper is of high interest to the journal’s readership. Congratulations to the authors for completing this work. It’s clear that the authors were thoughtful in their approach and took steps to conduct a high-quality review and meta-analysis – for example, by searching both the peer-reviewed literature and gray literature, assessing publication bias using funnel plots and Egger’s regression test, and conducting sensitivity analyses.

To help strengthen the manuscript, I’m sharing comments and suggested revision for the authors to consider.

Abstract

1. The authors state: “Although there were many individual studies in Ethiopia, their results are inconclusive…” It would be helpful for the authors to specify what the studies are about – e.g., “studies in Ethiopia about [specify here – e.g., nutritional status among adolescents]” in order to improve clarity.

2. The authors write, “The prevalence of overweight/obesity, stunting and thinness are higher in Ethiopian adolescents…” The authors could consider instead writing that the prevalence was “high,” since the authors do not compare the prevalence of these forms of malnutrition in Ethiopian adolescents to the prevalence in another population.

Introduction

3. It would strengthen the paper for the authors to define the double burden of malnutrition, given the many definitions in the literature. It would also be useful for the authors to explicitly state the level at which they are assessing the double burden of malnutrition, since the double burden can occur at the country, household, or individual levels.

4. Suggest re-wording the last sentence in the first paragraph for clarity. I believe the authors are referring to the idea that a continuum of care from childhood through adolescence is needed.

Methods

5. In the information sources and search strategy section, the authors state that “the reference lists of the retrieved studies were probed to collect articles that were not accessible through databases and search engines.” Were they accessible but just not picked up by the search strategy?

6. In the study selection section, the authors seem to describe how level of agreement was assessed for the title and abstract screening. I suggest they also describe how agreement was achieved for the full-text review.

7. In the study selection section, I suggest the authors describe how disagreements were resolved – e.g., through discussion.

8. In the data collection process, the authors mention that a “pilot test was conducted for all forms of using a representative sample of the studies to be reviewed…” What does it mean for the sample to be representative?

9. In the data collection process, the authors write that the study quality score was extracted from the full texts. Suggest deleting “study quality score” in the list of info that was extracted from the studies, since a study quality score cannot be extracted from the studies. It’s clear later on in the discussion section that the study quality scores were included in the extraction sheet.

10. I suggest the authors revise the quality appraisal and risk of bias section for clarity – e.g., the authors write that “the source of discrepancy was investigated through making a thorough revision.” Did the authors mean that disagreements were resolved through discussion?

Results

11. The first sentence states that there were a total of 1289 records, while the figure indicates that there were a total of 1269 records. Is this a typo?

12. Suggest Table 3 includes “(n)” at the top of the columns that report n’s.

13. The authors may want to consider moving some of the tables and/or figures to the supplementary materials section.

14. Figures 16, 17, and 18 are difficult to read. They may need to be re-formatted.

Discussion Conclusion

15. It would be helpful for the authors to discuss the quality of the studies in the discussion section.

16. Is the prevalence of overweight and obesity among Ethiopian adolescents increasingly rapidly? If so, what do the authors think about generating a pooled prevalence estimate based on studies conducted between 2010 – 2020?

17. The authors could consider moving the first paragraph of the discussion section to the introduction, as it describes the problem of malnutrition.

18. The authors compare their findings to results from other LMICs and high-income countries. The authors could consider also describing how the results compare to those from other countries in the region.

19. The authors write “The pooled estimates of overweight/obesity showed high increase” and then later refer to the “rising prevalence of overweight/obesity.” Since trends were not evaluated in this study, I suggest not using words that indicate an increasing trend in order to ensure that the findings support the conclusions.

20. The authors argue that surveillance of nutritional status is needed. Specifically, what would the authors recommend for improved surveillance? Some specific examples of recommendations could strengthen the manuscript.

Overall

21. I suggest proofreading the entire manuscript as there are some typos and sentences that could be revised for clarity.

22. References for some of the statements made, particularly in the introduction and discussion sections, are needed.

For example:

• “According to WHO, interest and focus on adolescent health and nutrition is relatively recent.”

• “They can also increase in weight as much as half of their adult body weight.”

• “Males are also more physically active than females.”

• “In developing countries including Ethiopia, girls usually stay at home due to cultural influence not to move from place to place than boys which results in physical inactivity and ultimately lead to overweight and obesity.”

• “Protected source of drinking water is a mechanism to prevent intestinal parasites and other communicable diseases which causes poor nutritional status.”

• “Unprotected water source results in repeated infections, depressed immunity increasing the severity and duration of diseases.”

23. The authors may want to reference some of the latest articles on the double burden of malnutrition published in the 2019 Lancet series on the double burden of malnutrition: https://www.thelancet.com/series/double-burden-malnutrition#:~:text=The%20double%20burden%20of%20malnutrition%20is%20the%20coexistence%20of%20overnutrition,community%2C%20household%2C%20and%20individual.

24. Since micronutrient deficiencies and associated morbidity and mortality are a persisting challenge in Ethiopia, it may strengthen the paper for the authors to discuss micronutrient deficiencies (even if briefly) in the paper (e.g., in the discussion section, the authors could consider noting that there are other indicators of nutritional status that are important to measure and address).

Reviewer #4: Dear Editor

My apologies for the delay in submitting my recommendation for this paper. Also, I wish to thank you for providing me this opportunity to review this paper which aims to evaluate the important issue of double burden of malnutrition in a developing nation like Ethiopia. Please find below my suggestions:

Introduction

In second paragraph, authors can give statistical figures suggesting how nutrition transition is increasing overnutrition in Ethiopia alongwith existing issue of undernutrition.

Spelling and grammatical recheck needs to be done e.g. in last paragraph 'this study aimed at determining the pooled prevalence malnutruition and associated factors among Ethiopian adolescents'.

Methods

Exclusion/Inclusion criteria should include the year of studies that were selected for the present manuscript.

7. PLOS authors have the option to publish the peer review history of their article (what does this mean?). If published, this will include your full peer review and any attached files.

Reviewer #3: No

Reviewer #4: No

---

## [Author Response · Author response to Decision Letter 1]

24 Dec 2022

We have addressed all comments given by the editors in abstract introduction, methods, result, discussion and conclusion part point by point.

---

## [Editor Report · Decision Letter 2]

13 Feb 2023

Double burden of malnutrition and associated factors among adolescent in Ethiopia: A systematic review and meta-analysis

PONE-D-21-00351R2

Dear Dr. Gezaw,

We’re pleased to inform you that your manuscript has been judged scientifically suitable for publication and will be formally accepted for publication once it meets all outstanding technical requirements.

This manuscript has improved much across the revisions.

A final important component that I encourage you to add before final publication is to consider the time component. You are pooling data from across a couple of decades, over which the prevalence of unhealthy weight is likely to have changed. It would be very important to provide some information on whether/how unhealthy weight has changed over time. Indeed, in the abstract you use the word "increase" but it's not clear over what period an increase has occurred. Please add an analysis of time trends.

Kind regards,

Solveig A. Cunningham, Ph.D.

Academic Editor

PLOS ONE

---

## [Editor Report · Acceptance letter]

17 Feb 2023

PONE-D-21-00351R2 

Double burden of malnutrition and associated factors among adolescent in Ethiopia: A systematic review and meta-analysis 

Dear Dr. Gezaw:

I'm pleased to inform you that your manuscript has been deemed suitable for publication in PLOS ONE. Congratulations! Your manuscript is now with our production department. 

Kind regards, 

on behalf of

Dr. PLOS Manuscript Reassignment 

Staff Editor

PLOS ONE